# Reinforcement Learning-Based Decentralized Safety Control for Constrained Interconnected Nonlinear Safety-Critical Systems

**DOI:** 10.3390/e25081158

**Published:** 2023-08-02

**Authors:** Chunbin Qin, Yinliang Wu, Jishi Zhang, Tianzeng Zhu

**Affiliations:** 1School of Artificial Intelligence, Henan University, Zhengzhou 450046, China; qcb@henu.edu.cn (C.Q.); 104754222776@henu.edu.cn (Y.W.); Kz520@henu.edu.cn (T.Z.); 2School of Software, Henan University, Kaifeng 475000, China

**Keywords:** interconnected nonlinear safety-critical systems, barrier function, asymmetric input constraints, safety constraints, decentralized control

## Abstract

This paper addresses the problem of decentralized safety control (DSC) of constrained interconnected nonlinear safety-critical systems under reinforcement learning strategies, where asymmetric input constraints and security constraints are considered. To begin with, improved performance functions associated with the actuator estimates for each auxiliary subsystem are constructed. Then, the decentralized control problem with security constraints and asymmetric input constraints is transformed into an equivalent decentralized control problem with asymmetric input constraints using the barrier function. This approach ensures that safety-critical systems operate and learn optimal DSC policies within their safe global domains. Then, the optimal control strategy is shown to ensure that the entire system is uniformly ultimately bounded (UUB). In addition, all signals in the closed-loop auxiliary subsystem, based on Lyapunov theory, are uniformly ultimately bounded, and the effectiveness of the designed method is verified by practical simulation.

## 1. Introduction

Over the past few decades, safety has received increasing attention in autonomous driving [1], intelligent robots [2], robotic arms [3], adaptive cruise control [4], etc. The design of these systems and controllers require that the system state trajectories evolve within a set called the safe set, reflecting the inherent properties of the system [5]. In practice, many engineering systems must operate within a specific safety range, beyond which the controlled system may be at risk [6]. Safety-critical systems primarily refer to systems having control behaviors that prioritize safety. The designed control schemes aim to reduce the potential for severe consequences, such as personal injury and environmental pollution, which may arise due to system shutdown or operational errors [7]. To ensure the safety and reliability of the system, scholars developed many safety control schemes. The classical approach focused on extending and applying Naguma’s theorem to safe sets defined by continuously differentiable functions [8]. In particular, barrier functions have become an effective tool for verifying security and have been widely used in [9,10,11]. They were used to convert a system with security constraints into an equivalency system that satisfies security requirements and then a security controller was designed to protect the system. In [9,10], penalty functions and BF-based state transitions were employed to merge states into a reinforcement learning framework to solve optimal control problems with full-state constraints. In [11], a safe non-strategic reinforcement learning method to solve secure nonlinear systems with dynamic uncertainty was proposed. In [12,13], a new secure reinforcement learning method was proposed to solve secure nonlinear systems with symmetric input constraints. However, the results in [9,10,11,12,13], mentioned above, were mainly based on studying the optimal safety control in a single continuous-time/discrete-time nonlinear system. The security control of interconnected systems has not been fully resolved.

On the other hand, interconnected systems consist of multiple subsystems with interconnected characteristics, and designing controllers for them through a concept similar to that of a single-system approach is difficult [14]. To solve this problem [15,16,17], the decentralized control approach, based on local subsystem information, was proposed. This approach involved using multiple controllers to control the interconnected systems. In [18,19], the decentralized control approach differed by initially decomposing the entire system control problem into a series of subproblems that could be solved independently. The solutions to the subproblems (i.e., independent controllers) were then joined to form a decentralized controller to stabilize the entire system. In addition, implementing the decentralized control algorithm used only the local subsystem’s knowledge, not the complete system’s information. Recently, scholars have proposed many schemes or techniques for designing decentralized controllers, including quantization techniques [20], fuzzy techniques [21], and optimal control methods [22]. This paper develops decentralized control strategies from the optimal control perspective. Problems of optimal control are usually solved via the solution of the Hamilton–Jacobi–Bellman (HJB) partial differentiation equation [23,24]. However, the HJB equation is generally not solvable analytically due to its inherent nonlinearity [25,26]. Therefore, adaptive dynamic programming (ADP) and reinforcement learning (RL) algorithms were proposed to obtain numerical solutions to the HJB equation and were widely applied to nonlinear interconnected systems [27,28,29,30]. In [31,32], the two previously mentioned algorithms could be deemed closely related, as they exhibited similar characteristics in addressing optimal control problems. For example, in [27,28], the distributed optimal controller was designed using robust ADP for nonlinear interconnected systems with unknown dynamics and parameters. In [29], the optimal decentralized control problem for interconnected nonlinear systems subject to stochastic dynamics was solved by enhancing the performance function of the auxiliary subsystem and transforming the original control problem into a set of optimal control strategies sampled in periodic patterns. Furthermore, in [30], the identifier–critic network framework was used to solve the problem of decentralized event-triggered control based on sliding-mode surfaces, avoiding the need for knowledge of the system’s internal dynamics. It is worth noting that the control results provided in [27,28,29,30] did not consider input constraints.

Control constraints are commonly encountered in industrial processes, where they are widespread and have a detrimental impact on the performance of systems [33,34]. Therefore, the study of constrained nonlinear systems is of practical importance. In [35,36], the RL-based decentralized algorithm was developed for tracking control of constrained interconnected nonlinear systems. In [37], the problem of decentralized optimal control of a constrained interconnected nonlinear system was solved by introducing a nonquadratic performance function to overcome the symmetric input constraint. The results in [35,36,37], mentioned above, mainly addressed the symmetric input constraint. However, the problem of asymmetric input constraints was identified in several project cases [38,39]. In [40], the optimal decentralized control problem with asymmetric input constraints was solved by designing a new non-quadratic performance function. In [41], a new performance function was proposed for interconnected nonlinear systems to successfully overcome the asymmetric input constraint and to solve the decentralized fault-tolerant control problem. However, none of the above studies considered the safety of the system. The optimal decentralized safety control (DSC) for constrained interconnected nonlinear safety-critical systems has not been thoroughly investigated thus far, which inspired our current study.

Motivated by previous discussions, this paper proposes an RL-based decentralized DSC strategy for constrained interconnected nonlinear safety-critical systems. The primary achievements are concluded below:The reinforcement learning algorithm is used to solve the optimal DSC problem for restricted interconnected nonlinear safety-critical systems, and the asymmetric input constraint is successfully solved. The method optimizes the control strategy by minimizing the performance function, ensuring the safety of the system’s state, while considering the asymmetric input constraints.Nonlinear interconnected safety-critical systems with asymmetric input constraints and safety constraints are converted to equivalent systems that satisfy user-defined safety constraints using barrier functions. Unlike the nonlinear safety-critical systems [3,9,10,13], this paper solves the security constraint problem of the interconnection term through the potential barrier function, which ensures the interconnected nonlinear safety-critical system satisfies the security constraint.The asymmetric input constraints are solved by utilizing a single CNN architecture for online approximation of the performance function. Theoretical demonstrations show that the optimal DSC method can achieve uniformly ultimately bounded (UUB) system states and neural network weight estimation errors. In addition, a simulation example verified the feasibility and effectiveness of the developed DSC method.

The remainder of this article is structured as follows. In Section 2, the issue formulation and conversion are presented. In Section 3, the decentralized optimal safety DSC design scheme is presented. The design scheme for the critical neural network is presented in Section 4. In Section 5, the analyses of system stability are presented. In Section 6, the simulation sample demonstrates the effectiveness of the presented approach. Lastly, conclusions are given in Section 7.

## 2. Preliminaries

### 2.1. Problem Descriptions

Consider a constrained interconnected nonlinear safety-critical system composed of n subsystems and the formula below: (1)$$\begin{array}{c} \begin{array}{c} \left\{\begin{array}{c} \dot{x_{i}}\left(t\right)=f_{i}\left(x_{i}\left(t\right)\right)+g_{i}\left(x_{i}\left(t\right)\right)u_{i}\left(t\right)+▵h_{i}\left(x\left(t\right)\right),\\ x_{i}\left(0\right)=x_{i0},\;i=1,2,\ldots ,n, \end{array}\right. \end{array} \end{array}$$
where $x_{i}\left(t\right)\in R^{n_{i}}$ is the *i*th subsystem’s state vector and $x_{i}\left(0\right)$ represents the initial state, $x=\left[x_{1}^{T},x_{2}^{T},\ldots ,x_{n}^{T}\right]\in R^{\sum_{i=1}^{n}n_{i}}$ represents the overall state vector of the constrained interconnected nonlinear safety-critical system, $u_{i}={\left[u_{i,1},u_{i,2},\;\ldots \;u_{i,j}\right]}^{T}\in k_{i}$ represents the control input, and the set of asymmetric constraints is represented as $k_{i}=\left\{u_{i,m_{i}}\in R^{m_{i}},h_{i\;min}\leq \left|u_{i,j}\right|\leq h_{i\;max},j=1,2,\ldots ,m_{i}\right\}$ with $h_{i\;min}$ and $h_{i\;max}$ being the asymmetric saturating minimum and maximum bounds, $f_{i}\left(\cdot \right)\in R^{n_{i}}$ and $g_{i}\left(\cdot \right)\in R^{n_{i}\times m_{{}_{i}}}$ represent the drift system dynamics and input dynamics of the *i*th subsystem, respectively, and are Lipschitz continuous, and $▵h_{i}\in R^{n_{i}}$ represents the unknown interconnected term.

To simplify the design of the controller, let us introduce some assumptions. For i=1,2,…,n, we suppose the equilibrium of the *i*th subsystem’s state is $x_{i}=0$.

**Assumption** **1.**
*For i=1,2,…,n, the $▵h_{i}\left(x\right)$ satisfies the below unmatched condition:*

$$△h_{i}=\eta_{i}\left(x_{i}\right)P_{i}\left(x\right),$$

*where $\eta_{i}\left(x_{i}\right)$ is a known function with $\eta_{i}\left(x_{i}\right)\in R^{n_{i}\times q_{{}_{i}}}\neq g_{i}\left(x_{i}\right)$, and $P_{i}\left(x\right)$ is a bounded vector function that satisfies*

(2)
$$∥P_{i}\left(x\right)∥\leq \sum_{j=1}^{n}b_{i,j}\beta_{i,j}\left(x_{j}\right),$$

*where $b_{i,j}>0$ is a constant, and $\beta_{i,j}\left(x_{j}\right)$ are normal definite functions. Furthermore, $\beta_{i,j}\left(0\right)=0$ and $P_{i}\left(0\right)=0$. Then, assuming $\beta_{j}\left(x_{j}\right)=max_{1\leq i\leq n}\left\{\beta_{i,j}\left(x_{j}\right)\right\}$, the unequal Equation (Equation 2) is denoted as:*

(3)
$$∥P_{i}\left(x\right)∥\leq \sum_{j=1}^{n}C_{i,j}\beta_{j}\left(x_{j}\right),$$

*where $C_{i,j}\geq \left(b_{i,j}\beta_{i,j}\left(x_{j}\right)\right)/\beta_{j}\left(x_{j}\right)$ is a positive constant, and j=1,2,…,n.*


**Remark** **1.**
*It is noted that constraints (Equation 2) and (Equation 3) specified by Assumption 1 are strict restrictions on specific interrelated nonlinear systems. Nevertheless, when we consider the function $P_{i}\left(x\right)$ that satisfies no constraints (Equation 2) and (Equation 3), we discover that the calculational costs to address the stability of the closed-loop system are high. In fact, in real-world applications, constraints like inequalities (Equation 2) and (Equation 3) impose on the mismatched interconnection terms of the system (Equation 1) [40,42].*


**Assumption** **2.**
*For i=1,2,…,n, the known function $g_{i}\left(x_{i}\right)$ is bounded as $∥g_{i}\left(x_{i}\right)∥\leq g_{i,m}$, where $g_{i,m}$ is a known constant. Furthermore, $rank\left(g_{i}\left(x_{i}\right)\right)=m_{i}$ and $g_{i}^{T}\left(x_{i}\right)\eta_{i}\left(x_{i}\right)=0$.*


Based on the *i*th subsystem (Equation 1) described, the *i*th auxiliary subsystem is designed as: (4)$$\dot{x_{i}}=f_{i}\left(x_{i}\right)+g_{i}\left(x_{i}\right)u_{i}+\left(I_{n_{i}}-g_{i}\left(x_{i}\right)g_{i}^{+}\left(x_{i}\right)\right)\eta_{i}\left(x_{i}\right)v_{i},$$
where $v_{i}\in R^{q_{i}}$ is used to compensate for mismatched interconnections and stands for auxiliary control, $g_{i}^{+}\left(x_{i}\right)\in R^{m_{i}\times n_{i}}$ is Moore–Penrose pseudo-reverse. According to Assumption 2, it can be found that the matrix $g_{i}^{+}\left(x_{i}\right)={\left(g_{i}^{T}\left(x_{i}\right)g_{i}\left(x_{i}\right)\right)}^{-1}g_{i}^{T}\left(x_{i}\right)$ and $g_{i}^{+}\left(x_{i}\right)\eta_{i}\left(x_{i}\right)={\left(g_{i}^{T}\left(x_{i}\right)g_{i}\left(x_{i}\right)\right)}^{-1}g_{i}^{T}\left(x_{i}\right)\eta_{i}\left(x_{i}\right)=0$. Then, we rewrite the auxiliary subsystem (Equation 4) as: (5)$$\dot{x_{i}}=f_{i}\left(x_{i}\right)+g_{i}\left(x_{i}\right)u_{i}+\eta_{i}\left(x_{i}\right)v_{i}.$$

### 2.2. Security Conversion Issues

For the *i*th subsystem in the system (Equation 1), its state $x_{i}={\left[x_{i,1},x_{i,2},\ldots ,x_{i,k}\right]}^{T}$ satisfies the following security constraints: (6)$$\begin{array}{c} \begin{array}{c} \left\{\begin{array}{c} x_{i,1}\in \left(a_{i,1},A_{i,1}\right),\\ x_{i,2}\in \left(a_{i,2},A_{i,2}\right),\\ \;\;.\\ \;\;.\\ \;\;.\\ x_{i,k}\in \left(a_{i,k},A_{i,k}\right). \end{array}\right. \end{array} \end{array}$$
For nonlinear interconnect safety-critical systems with asymmetric input constraints and security constraints, we need to define the performance function as: (7)$$J_{i}\left(x_{i}\right)=\int_{t}^{\infty }e^{-\alpha_{i}\left(\tau -t\right)}\left(\iota_{i}+\Theta \left(x_{i},u_{i},v_{i}\right)\right)d\tau ,$$
where $\alpha_{i}$ is the discount factor, $\iota_{i}\left(x_{i}\right)=h_{i}\beta_{j}^{2}\left(x_{i}\right)$ and $\Theta \left(x_{i},u_{i},v_{i}\right)=x_{i}^{T}H_{i}x_{i}+W_{i}\left(u_{i}\right)+\xi_{i}v_{i}^{T}v_{i}$ with $H_{i}$ and $W_{i}\left(u_{i}\right)$ are positive definite functions, where $h_{i}$ and $\xi_{i}$ are positive design parameters.

**Remark** **2.**
*Due to accounting for safety constraints and asymmetric input constraints in (Equation 7), the optimal control law does not converge to zero while the system state achieves the stable phase [43]. The discount factor $\alpha_{i}=0$, $J_{i}\left(x_{i}\right)$ may be unbounded, so it is necessary to consider the discount factor.*


**Problem** **1.**
*(Decentralized control problems with security constraints and asymmetric input constraints) Consider the safety-critical system (Equation 1) and find the policy $u_{i}\left(.\right)$ and auxiliary control strategy $v_{i}\left(.\right):R^{n_{i}}\to R^{m_{i}}$ in the ith subsystem. The performance function is given by (Equation 7) with the ith subsystem state $x_{i}={\left[x_{i,1},\ldots ,x_{i,k}\right]}^{T}$ and the control input $u_{i}$ satisfying the following conditions:*

(8)
$$\begin{array}{cc} u_{i,min} & \leq u_{i,j}\leq u_{i,max},\;\left|u_{i,min}\right|\neq \left|u_{i,max}\right|, \end{array}$$


(9)
$$x_{i,k}\in \left(a_{i,k},A_{i,k}\right),\;\forall k=1,\ldots ,n_{i}.$$



Ensure that the security-critical system state is consistently within the security constraints. Further, the definitions of some barrier functions are given.

**Definition** **1**(Barrier function [9,10])**.**
*The function $B\left(\cdot \right):R\to R$ defined on interval (a, A) is referred to as the barrier function if*
(10)$$B\left(z;a,A\right)=\log_{}\frac{A\left(a-z\right)}{a\left(A-z\right)},\;\forall z\in \left(a,A\right),$$
where *a* and *A* are two constants satisfying a<A. Moreover, the potential function is invertible on the interval (a,A), i.e.,
(11)$$B^{-1}\left(y;a,A\right)=\frac{aA\left(e^{\frac{y}{2}}-e^{-\frac{y}{2}}\right)}{ae^{\frac{y}{2}}-Ae^{-\frac{y}{2}}},\;\forall y\in R.$$
Furthermore, the derivative of (Equation 11) is
(12)$$\frac{dB^{-1}\left(y;a,A\right)}{dy}=\frac{Aa^{2}-aA^{2}}{a^{2}e^{y}-2aA+A^{2}e^{-y}}.$$

Based on Definition 1, we consider the state transition based on the potential barrier function as follows: (13)$$\begin{array}{c} s_{i,k}=B\left(x_{i,k};a_{i,k},A_{i,k}\right), \end{array}$$
(14)$$x_{i,k}=B^{-1}\left(s_{i,k};a_{i,k},A_{i,k}\right),$$
where $k=1,2,\ldots ,n_{i}$. So, the $x_{i,k}$’s derivative concerning *t* is $\frac{dx_{i,k}}{dt}=\frac{dx_{i,k}}{ds_{i,k}}\frac{ds_{i,k}}{dt}$, and after using Definition 1, we obtain: $$\begin{array}{cc} {\dot{s}}_{i,k}= & \frac{a_{i,k+1}A_{i,k+1}\left(e^{\frac{s_{i,k+1}}{2}}-e^{-\frac{s_{i,k+1}}{2}}\right)}{a_{i,k+1}e\frac{s_{i,k+1}}{2}-A_{i,k+1}e-\frac{s_{i,k+1}}{2}}\times \frac{A_{i,k}^{2}e^{-s_{i,k}}-2a_{i,k}A_{i,k}+a_{i,k}^{2}e^{s_{i,k}}}{A_{i,k}a_{i,k}^{2}-a_{i,k}A_{i,k}^{2}}\\ = & F_{ik}\left(s_{i,k},s_{i,k+1}\right),\;k=1,\;\ldots ,\;n_{i}-1,\\ {\dot{s}}_{i,n_{i}}= & {\dot{x}}_{i}\times \frac{A_{i,k}^{2}e^{-s_{i,k}}-2a_{i,k}A_{i,k}+a_{i,k}^{2}e^{s_{i,k}}}{A_{i,k}a_{i,k}^{2}-a_{i,k}A_{i,k}^{2}}\\ = & F_{i,n_{i}}\left(s_{i,n_{i}}\right)+G_{i,n_{i}}\left(s_{i,n_{i}}\right)u_{i,n_{i}}+Y_{i,n_{i}}\left(s_{i,n_{i}}\right), \end{array}$$
where
$$\begin{array}{cc} F_{i,n_{i}}\left(s_{i}\right) & =\frac{a_{i,n_{i}}^{2}e_{i,n_{i}}^{s_{i}}-2a_{i,n_{i}}A_{i,n}+A_{i,n_{i}}e^{-s_{i,n_{i}}}}{A_{i,n_{i}}a_{i,n_{i}}^{2}-a_{i,n_{i}}A_{i,n_{i}}^{2}}\times f_{i}\left(\left[B_{i,1}^{-1}\left(s_{i,1}\right)\ldots B_{i,n_{i}}^{-1}\left(s_{i,n_{i}}\right)\right]\right),\\ G_{i,n_{i}}\left(s_{i}\right) & =\frac{a_{i,n_{i}}^{2}e_{i,n_{i}}^{s_{i}}-2a_{i,n_{i}}A_{i,n_{i}}+A_{i,n_{i}}e^{-s_{i,n_{i}}}}{A_{i,}a_{i,n_{i}}^{2}-a_{i,n_{i}}A_{i,n_{i}}^{2}}\times \;g_{i}\left(\left[B_{i,1}^{-1}\left(s_{i,1}\right)\ldots B_{i,n_{i}}^{-1}\left(s_{i,n_{i}}\right)\right]\right), \end{array}$$
and $Y_{i,n_{i}}\left(s_{i,n_{i}}\right)$ is the interconnection term of the $n_{i}$th term in the *i*th subsystem.

Then, the interconnected nonlinear safety-critical system (Equation 1) can be rewritten as: (15)$$\dot{s_{i}}=F_{i}\left(s_{i}\right)+G_{i}\left(s_{i}\right)u_{i}\left(t\right)+Y_{i}\left(s_{i}\right),$$
where $F_{i}\left(s_{i}\right)={\left[F_{i1}\left(s_{i,1},s_{i,2}\right),\ldots ,F_{i,n_{i}}\left(s_{i}\right)\right]}^{T}$, $G_{i}\left(s_{i}\right)={\left[0,\ldots ,G_{i,n_{i}}\left(s_{i}\right)\right]}^{T}$ and $Y_{i}\left(s_{i}\right)$ is the unknown interconnected term.

Based on Assumption 1, we define the unknown interconnection term after the system transformation as: (16)$$Y_{i}\left(s_{i}\right)={℘}_{i}\left(s_{i}\right)U_{i}\left(s\right),$$
where ${℘}_{i}\left(s_{i}\right)={\left[{℘}_{1,n_{1}}\left(s_{1}\right),0,\ldots ,0\right]}^{T}$, and
$${℘}_{1,n_{1}}\left(s_{1}\right)=\frac{a_{i,2}^{2}e^{s_{i,2}}-2a_{i,2}A_{i,2}+A_{i,2}e^{-s_{i,2}}}{A_{i,2}a_{i,2}^{2}-a_{2}A_{i,2}^{2}}\times \eta_{1,n_{1}}\left(x_{1}\right),$$
and $U_{i}\left(s_{i}\right)$ is a bounded vector function that satisfies
(17)$$\begin{array}{c} ∥U_{i}\left(s\right)∥\leq \sum_{j=1}^{n}b_{i,j}\vartheta_{i,j}\left(s_{j}\right), \end{array}$$
where $\vartheta_{i,j}\left(s_{j}\right)$ is a positive definite function. Then, assuming $\vartheta_{j}\left(s_{j}\right)=max_{1\leq i\leq n}\left\{\vartheta_{i,j}\left(s_{j}\right)\right\}$ and $\vartheta_{j}\left(s_{j}\right)={\left[\vartheta_{j,1}\left(s_{j,1},s_{j,2}\right),\ldots ,\vartheta_{j,n_{i}}\left(s_{j}\right)\right]}^{T}$, where
(18)$$\vartheta_{j,n_{i}}\left(s_{j}\right)=\frac{a_{j,n_{i}}^{2}e_{j,n_{i}}^{s_{j}}-2a_{j,n_{i}}A_{j,n_{i}}+A_{j,n_{i}}e^{-s_{j,n_{i}}}}{A_{j,n_{i}}a_{j,n_{i}}^{2}-a_{j,n_{j}}A_{j,n_{i}}^{2}}\times \beta_{j}\left(\left[B_{j,1}^{-1}\left(x_{1}\right)\ldots B_{j,n_{i}}^{-1}\left(x_{j}\right)\right]\right).$$
According to (Equation 3) and (Equation 18), the inequality (Equation 17) is expressed as: (19)$$∥U_{i}\left(s\right)∥\leq \sum_{j=1}^{n}S_{i,j}\vartheta_{j}\left(s_{j}\right),$$
where $S_{i,j}\geq \left(b_{i,j}\vartheta_{i,j}\left(s_{j}\right)\right)/\vartheta_{j}\left(s_{j}\right)$ is a positive constant, and i,j=1,2,…,n.

**Assumption** **3.**
*$F_{i}\left(s_{i}\right)$ is Lipschitz continuous with $F_{i}\left(0\right)=0$, $P_{i}\left(0\right)=0$, $G_{i}\left(s_{i}\right)$ and ${℘}_{i}\left(s_{i}\right)$ are upper-bounded, then $∥F_{i}\left(s_{i}\right)∥\leq f_{i,m_{i}}∥s_{i}∥$, $∥G_{i}\left(s_{i}\right)∥\leq g_{i,m_{i}}$, and $∥{℘}_{i}\left(s_{i}\right)∥\leq \eta_{i,m_{i}}$, $∥U_{i}\left(s_{i}\right)∥\leq P_{i,m_{i}}∥s_{i}∥$, where $f_{i,m_{i}}$, $g_{i,m_{i}}$, $\eta_{i,m_{i}}$, $P_{i,m_{i}}$ are positive constants. $rank\left(G_{i}\left(s_{i}\right)\right)=m_{i}$ and $G_{i}^{T}\left(s_{i}\right){℘}_{i}\left(s_{i}\right)=0$. Moreover, the modified system (Equation 15) is within the manageable range, and $s_{i}=0$ is the balance point for (Equation 15).*


**Lemma** **1**([32])**.**
*$\forall \left(s_{1},s_{2}\right)\in R^{2}$, we have the following condition,*
$$\begin{array}{c} s_{1}s_{2}\leq \frac{\varepsilon_{1}{\;}^{p_{1}}}{p_{1}}|s_{1}{|}^{p_{1}}+\frac{1}{p_{2}\varepsilon_{1}{\;}^{p_{}2}}{\left|s_{2}\right|}^{p_{2}}, \end{array}$$
*where $\varepsilon_{1}>0,\left(p_{1}-1\right)\left(p_{2}-1\right)=1$ and $p_{1},p_{2}>1$.*

**Remark** **3.**
*The barrier function in Definition 1, which has the following characteristics, ensures that the safety-critical system (Equation 15) always satisfies the safety constraints [9,10].*

*1.* 
*The state $s_{i}$ of the system is restricted to be bounded, so the system state $x_{i}$ satisfies constraints (Equation 8) and (Equation 9), i.e.,*

$$\begin{array}{c} \left|B(z_{i};a_{i},A_{i})\right|<+\infty ,\;\forall z_{i}\in \left(a_{i},A_{i}\right). \end{array}$$

*2.* 
*When the system’s state approaches the boundary of the safety area, the barrier function changes as follows:*

$$\begin{array}{c} \lim_{z_{i}\to a_{i}^{+}}B\left(z_{i};a_{i},A_{i}\right)=-\infty ,\\ \lim_{z_{i}\to A_{i}^{-}}B\left(z_{i};a_{i},A_{i}\right)=+\infty . \end{array}$$

*3.* 
*The barrier function fails to function when the system state reaches equilibrium, i.e.,*

$$\begin{array}{c} B\left(0;a_{i},A_{i}\right)=0,\;\forall a_{i}<A_{i}. \end{array}$$




## 3. Decentralized Optimal DSC Design

This section consists of two main subsections to establish the decentralized optimal DSC method. First, the security constraint problem is dealt with through the systematic transformation of the barrier function and the HJB equation for the *i*th auxiliary subsystem without security constraints is developed by introducing the improved performance function. Finally, the decentralized safety controller is constructed by solving the HJB equation for the auxiliary subsystem.

### 3.1. Barrier Function Conversion

According to the *i*th subsystem (Equation 15) described, the *i*th auxiliary subsystem is designed as: (20)$$\dot{s_{i}}=F_{i}\left(s_{i}\right)+G_{i}\left(s_{i}\right)u_{i}+\left(I_{n_{i}}-G_{i}\left(s_{i}\right)G_{i}^{+}\left(s_{i}\right)\right){℘}_{i}\left(s_{i}\right)v_{i},$$
where $G_{i}^{+}\left(s_{i}\right)\in R^{m_{i}\times n_{i}}$ is Moore–Penrose pseudo-reverse. According to Assumptions 2 and 3, the matrix if found to be $G_{i}^{+}\left(s_{i}\right)={\left(G_{i}^{T}\left(s_{i}\right)G_{i}\left(s_{i}\right)\right)}^{-1}G_{i}^{T}\left(s_{i}\right)$ and $G_{i}^{+}\left(s_{i}\right){℘}_{i}\left(s_{i}\right)={\left(G_{i}^{T}\left(s_{i}\right)G_{i}\left(s_{i}\right)\right)}^{-1}G_{i}^{T}\left(s_{i}\right){℘}_{i}\left(s_{i}\right)=0$. Then, the auxiliary subsystem (Equation 20) is rewritten as: (21)$$\dot{s_{i}}=F_{i}\left(s_{i}\right)+G_{i}\left(s_{i}\right)u_{i}+{℘}_{i}\left(s_{i}\right)v_{i}.$$

Regarding the converted system (Equation 15), analogous to (Equation 7), the performance function below is introduced: (22)$$V_{i}\left(s_{i}\right)=\int_{t}^{\infty }e^{-\alpha_{i}\left(\tau -t\right)}\left(\pi_{i}+\gamma \left(s_{i},u_{i},v_{i}\right)\right)d\tau ,$$
where $\pi_{i}\left(s_{i}\right)=h_{i}\vartheta_{j}^{2}\left(s_{i}\right)$ and $\gamma \left(s_{i},u_{i},v_{i}\right)=s_{i}^{T}Q_{i}s_{i}+W_{i}\left(u_{i}\right)+\xi_{i}v_{i}^{T}v_{i}$, $Q_{i}$ is the positive definition matrix. Furthermore, $s_{i0}=s_{i}\left(0\right)$ denotes the initial state, and $W_{i}\left(u_{i}\right)$ is a non-quadratic utility function that solves the asymmetric input constraint. Then, $W_{i}\left(u_{i}\right)$ is defined in the following form: (23)$$W_{i}\left(u_{i}\right)=\sum_{j=1}^{m_{i}}2\lambda_{i}\int_{c_{i}}^{u_{i,j}}\Psi^{-1}\left(\left(v_{i}-c_{i}\right)/\lambda_{i}\right)dv_{i},$$
where $\lambda_{i}=\left(h_{i\;max}-h_{i\;min}\right)/2$ and $c_{i}=\left(h_{i\;max}+h_{i\;min}\right)/2$, and $\Psi_{i}\left(.\right)$ represent the monotonic odd function, where $\Psi_{i}\left(0\right)=0$. In this paper, without sacrificing generality, $\Psi_{i}\left(s_{i}\right)=\left(e^{s_{i}}-e^{-s_{i}}\right)/\left(e^{s_{i}}+e^{-s_{i}}\right)$.

**Remark** **4.**
*Unlike the traditional form of symmetric input constraints [35], this article considered asymmetric constraints on the controlling inputs [44]. The revised hyperbolic tangent function presented in (Equation 22) effectively transforms the asymmetric constrained control problem into an unconstrained control problem by devising different maximum and minimum bounds.*


**Problem** **2.**
*(Optimal decentralized control problems with asymmetric input constraints) Finding the control policy $u_{i}$ and auxiliary control strategy $v_{i}$ in the ith subsystem, the performance function becomes (Equation 22).*


Based on the subsystem (Equation 21), as well as the performance function (Equation 22), the corresponding Hamiltonian is given by: (24)$$\begin{array}{cc} H(s_{i},u_{i},v_{i},\nabla V_{i}\left(s_{i}\right))=\; & {\left(\nabla V_{i}\left(s_{i}\right)\right)}^{T}\left(F_{i}\left(s_{i}\right)+G_{i}\left(s_{i}\right)u_{i}\left(t\right)+{℘}_{i}\left(s_{i}\right)v_{i}\right)\\ \; & +\pi_{i}+\gamma \left(s_{i},u_{i},v_{i}\right)-\alpha_{i}V_{i}, \end{array}$$
with $\nabla V_{i}\left(s_{i}\right)=\frac{\partial V_{i}\left(s_{i}\right)}{\partial s_{i}}$.

The optimal performance function is
(25)$$V_{i}^{*}\left(s_{i}\right)=\min_{u_{i},v_{i}\in \Psi \left(\Omega_{i}\right)}V_{i}\left(s_{i}\right),$$
where $\Psi (\Omega_{i})$ is a collection of all acceptable control policies and auxiliary control strategies for $\Omega_{i}$.

Based on Bellman’s optimality principle [31], $V_{i}^{*}\left(s_{i}\right)$ in (Equation 25) satisfies the HJB
(26)$$\min_{u_{i},v_{i}\in \Psi \left(\Omega_{i}\right)}H\left(s_{i},u_{i},v_{i},\nabla V_{i}^{*}\left(s_{i}\right)\right)=0,$$
where $\nabla V_{i}^{*}\left(s_{i}\right)=\frac{\partial V_{i}^{*}\left(s_{i}\right)}{\partial s_{i}}$. Then, the optimal control policy and the auxiliary control policy can be derived as follows: (27)$$u_{i}^{*}\left(s_{i}\right)=-\lambda_{i}\tanh\left(\frac{1}{2\lambda_{i}}G_{i}^{T}\left(s_{i}\right)\nabla V_{i}^{*}\left(s_{i}\right)\right)+c_{i},$$
(28)$$v_{i}^{*}\left(s_{i}\right)=-\frac{1}{2\xi_{i}}{℘}_{i}^{T}\nabla V_{i}^{*}\left(s_{i}\right),$$
where $c_{i}=\left[c_{1},\ldots ,c_{m_{i}}\right]$.

Substituting $u_{i}^{*}\left(s_{i}\right)$ and $v_{i}^{*}\left(s_{i}\right)$ into (Equation 26), the HJB equation is rewritten as: (29)$$\begin{array}{cc} {\left(\nabla V_{i}^{*}\left(s_{i}\right)\right)}^{T}F_{i}\left(s_{i}\right)+{\left(\nabla V_{i}^{*}\left(s_{i}\right)\right)}^{T}G_{i}\left(s_{i}\right)u_{i}^{*}\left(s_{i}\right)- & \xi_{i}{∥v_{i}^{*}\left(s_{i}\right)∥}^{2}\\ -\alpha_{i}V_{i}^{*}+\pi_{i}\left(s_{i}\right)+s_{i}^{T}Q_{i}s_{i}+W_{i}\left(u_{i}^{*}\left(s_{i}\right)\right) & =0, \end{array}$$
with $V_{i}^{*}\left(0\right)=0$.

Through the BF-based system transformation, the decentralized control problem 1 with asymmetric input constraints and security constraints is transformed into an unconstrained optimization problem, i.e., the decentralized control problem 2. Next, the following lemma is discussed to ensure the equivalence between the decentralized control problems 1 and 2.

**Lemma** **2.**
*Assume that Assumptions 1 to 3 are met and that control policy $u_{i}\left(\cdot \right)$ and auxiliary control strategy $v_{i}\left(\cdot \right)$ solve the decentralized control problem 2 of (Equation 21). It follows, then, that the below holds:*

*1.* 
*If the initial state $x_{0}$ of the interconnected nonlinear safety-critical system (Equation 1) is in the range ($a_{i,k}$, $A_{i,k}$), $\forall k=1,2,\ldots ,n_{i}$, then the closed-loop system satisfies (Equation 6).*
*2.* 
*If the functions $H_{i}\left(x\right)$ and $Q_{i}\left(x\right)$ satisfies the condition $H_{i}\left(x_{i}\right)=Q_{i}\left(B_{i}\left(x_{i}\right)\right)=Q_{i}\left(s_{i}\right)$, the performance described in (Equation 22) is equivalent to the one in (Equation 7).*



**Proof.** Both the performance function and Assumption 3 satisfy the observability of zero states, guaranteeing the presence of the safety-optimal performance function $V_{i}^{*}\left(s_{i}\right)$. From (Equation 24), we obtain $\nabla V_{i}^{*}\left(t\right)\leq 0$, which allows us to obtain $V_{i}^{*}\left(s_{i}\left(t\right)\right)\leq V_{i}^{*}\left(s_{i}\left(0\right)\right)$ for all t≥0. Consequently, as stated in Remark 3, if the initial state $x_{i}\left(0\right)$ of the system (Equation 21) satisfies the security constraint (Equation 6), and $V_{i}^{*}\left(s_{i}\left(0\right)\right)$ is bounded, then the $V_{i}^{*}\left(s_{i}\left(t\right)\right)$ is also bounded. Finally, we obtain
(30)$$x_{i,k}\left(t\right)\in \left(a_{i,k},A_{i,k}\right),\;k=1,2,\ldots ,n_{i}.$$
Therefore, the given $u_{i}^{*}$ and $v_{i}^{*}$ satisfy the constraints of the decentralized control problem 1.Now, consider the state transition based on the barrier function described in (Equation 13) and (Equation 14). Since $x_{i}$ satisfies the constraints given in (Equation 8), each element of the state $s_{i}={\left[B_{i,1}\left(x_{i,1}\right),\ldots ,B_{i,k}\left(x_{i,k}\right)\right]}^{T}$ is finite. By comparing the performance functions (Equation 7) and (Equation 22), the equivalence relation $J_{i}\left(x_{i}\left(0\right)\right)=V_{i}\left(s_{i}\left(0\right)\right)$ is obtained, provided that $H_{i}\left(x_{i}\right)=Q_{i}\left(s_{i}\right)$. This completes the proof. □

### 3.2. Designing the Optimal DSC Strategy by Solving n HJB Equations

Throughout this section, we show that the optimal DSC strategies for interconnected nonlinear systems can be constructed by solving the *n* HJB equations.

**Theorem** **1.**
*Consider n subsystems under Assumptions 1 to 3 with DSC policies $u_{i}^{*}\left(s_{i}\right)$ and auxiliary control strategies $v_{i}^{*}\left(s_{i}\right)$, having the corresponding conditions as below:*

(31)
$${∥v_{i}^{*}\left(s_{i}\right)∥}^{2}<s_{i}^{T}Q_{i}s_{i}\;,t\geq t_{0}.$$

*Next, consider n positive constants $h_{i}^{*},\;i=1,2,\ldots ,n$, so that for anything $h_{i}\geq h_{i}^{*}$, the optimal DSC policies $u_{1}^{*}\left(s_{1}\right)$, $u_{2}^{*}\left(s_{2}\right)$, …, $u_{n}^{*}\left(s_{n}\right)$ guarantee that the interconnected nonlinear system (15) with security constraints is UUB.*


**Proof.** The Lyapunov candidacy function $L_{i,1}\left(s\right)$ below was selected:
(32)$$L_{i,1}\left(s\right)=\sum_{i=1}^{n}V_{i}^{*}\left(s_{i}\right),$$
where the $V_{i}^{*}\left(s_{i}\right)$ is defined in the same way as (Equation 22), and we denote the time derivative along the trajectory $\dot{s_{i}}=F_{i}\left(s_{i}\right)+G_{i}\left(s_{i}\right)u_{i}\left(t\right)+Y_{i}\left(s_{i}\right)$ as:
(33)$${\dot{L}}_{i,1}\left(s\right)=\sum_{i=1}^{n}{\left(\nabla V_{i}^{*}\right)}^{T}\left(G_{i}\left(s_{i}\right)u_{i}^{*}+F_{i}\left(s_{i}\right)+Y_{i}\left(s\right)\right).$$By using (Equation 27) and (Equation 28), we obtain:
(34)$${\left(\nabla V_{i}^{*}\left(s_{i}\right)\right)}^{T}G_{i}\left(s_{i}\right)=-2\lambda_{i}\tanh^{-T}\left(\frac{u_{i}^{*}-c_{i}}{\lambda_{i}}\right),$$
(35)$${\left(\nabla V_{i}^{*}\left(s_{i}\right)\right)}^{T}{℘}_{i}\left(s_{i}\right)=-2\xi_{i}{\left(v_{i}^{*}\left(s_{i}\right)\right)}^{T}.$$Inserting (Equation 29), (Equation 34) and (Equation 35) into (Equation 33), we have
(36)$$\begin{array}{c} {\dot{L}}_{i,1}\left(s\right)=\sum_{i=1}^{n}\left[\alpha_{i}V_{i}^{*}-\pi_{i}\left(s_{i}\right)-s_{i}^{T}Q_{i}s_{i}-W_{i}\left(u_{i}^{*}\right)+\xi_{i}{∥v_{i}^{*}\left(s_{i}\right)∥}^{2}-2\xi_{i}{\left(v_{i}^{*}\left(s_{i}\right)\right)}^{T}U_{i}\left(s\right)\right]. \end{array}$$According to the optimal DSC policy (Equation 27), the term $W_{i}\left(u_{i}^{*}\right)$ becomes
(37)$$W_{i}\left(u_{i}^{*}\left(s_{i}\right)\right)=2\lambda_{i}\sum_{j=1}^{m_{j}}\int_{0}^{u_{i,j}^{*}-c_{i}}\tanh^{-1}\left(\frac{u_{i}-c_{i}}{\lambda_{i}}\right)d\left(u_{i}-c_{i}\right).$$By appealing to the proof in [44], Equation (Equation 37) can be further reduced to
(38)$$\begin{array}{cc} \; & W_{i}\left(u_{i}^{*}\left(s_{i}\right)\right)=\underset{\beta_{1}}{\underbrace{\lambda_{i}^{2}\sum_{i=1}^{m_{i}}\left(\tanh^{-1}\left(\frac{u_{i,j}^{*}-c_{i}}{\lambda_{i}}\right)\right)}}\\ \; & -\underset{\beta_{2}}{\underbrace{2\lambda_{i}^{2}\sum_{j=1}^{m_{i}}\int_{0}^{\tanh^{-1}\left(\frac{u_{i,j}^{*}-c_{i}}{\lambda_{i}}\right)}\left(u_{i}-c_{i}\right)\tanh^{2}\left(u_{i}-c_{i}\right)d\left(u_{i}-c_{i}\right)}}, \end{array}$$
replacing (Equation 38) into (Equation 36), one has
(39)$$\begin{array}{cc} {\dot{L}}_{i,1}\left(s\right)\leq & -\sum_{i=1}^{n}\left(2\xi_{i}\left(s_{i}^{T}Q_{i}s_{i}-{∥v_{i}^{*}\left(s_{i}\right)∥}^{2}\right)\right)-\sum_{i=1}^{n}\left(1-2\xi_{i}\right)\left(s_{i}^{T}Q_{i}s_{i}\right)-\sum_{i=1}^{n}(\pi_{i}\left(s_{i}\right)\\ \; & -2\xi_{i}\sum_{j=1}^{m_{i}}∥v_{i}^{*}\left(s_{i}\right)∥b_{i,j}\vartheta_{i,j}\left(s_{j}\right)+\xi^{2}{∥v_{i}^{*}\left(s_{i}\right)∥}^{2})+\alpha_{i}V_{i}^{*}-\beta_{1}+\beta_{2}. \end{array}$$It is known from [45] that there is a positive constant $\delta_{i,M}$ such that $0\leq ∥\nabla V_{i}^{*}\left(s_{i}\right)∥\leq \delta_{i,M}$. Therefore, using Lemma 1, Assumption 1, (Equation 17), (Equation 19), and (Equation 27), we obtain
(40)$$\begin{array}{cc} 2\beta_{1} & \leq 2\lambda_{i}^{2}\tanh^{-T}\left(\frac{u_{i,j}^{*}-c_{i}}{\lambda_{i}}\right)\tanh^{-1}\left(\frac{u_{i,j}^{*}-c_{i}}{\lambda_{i}}\right)\\ \; & =\frac{1}{2}{\left(\nabla V_{i}^{*}\left(s_{i}\right)\right)}^{T}G_{i}\left(s_{i}\right)G_{i}^{T}\left(s_{i}\right)\left(\nabla V_{i}^{*}\left(s_{i}\right)\right)\\ \; & \leq \frac{1}{2}G_{i,m}^{2}\delta_{i,m}^{2}, \end{array}$$
Utilizing the integral median theorem [46] and the inequality (Equation 40), the $\beta_{2}$ (Equation 38) can be deduced as:
(41)$$\begin{array}{cc} \beta_{2} & =2\lambda_{i}^{2}\sum_{j=1}^{m_{i}}\tanh^{-1}\left(\frac{u_{i,j}^{*}-c_{i}}{\lambda_{i}}\right)\varpi_{i}\tanh^{-2}\varpi_{i}\\ \; & \leq 2\lambda_{i}^{2}\sum_{j=1}^{m_{i}}\tanh^{-1}\left(\frac{u_{i,j}^{*}-c_{i}}{\lambda_{i}}\right)\varpi_{i}\\ \; & \leq 2\lambda_{i}^{2}\tanh^{-T}\left(\frac{u_{i,j}^{*}-c_{i}}{\lambda_{i}}\right)\tanh^{-1}\left(\frac{u_{i,j}^{*}-c_{i}}{\lambda_{i}}\right)\\ & \leq \frac{1}{2}G_{i,m}^{2}\delta_{i,m}^{2}, \end{array}$$
where $\varpi_{i}\in \left(0,\tanh^{-1}\left(\frac{u_{i,j}^{*}-c_{i}}{\lambda_{i}}\right)\right)$.From [27], we conclude that $∥\alpha_{i}V_{i}^{*}\left(s_{i}\right)∥\leq \varrho_{i,m}$, where $\varrho_{i,m}$ is a positive constant. Then, plugging (Equation 40) and (Equation 41) into (Equation 39), and taking into consideration the conclusion mentioned above, we can rephrase inequality (Equation 39) as follows:
(42)$$\begin{array}{cc} \; & {\dot{L}}_{i,1}\left(s\right)\leq -\sum_{i=1}^{n}\left(2\xi_{i}\left(s_{i}^{T}Q_{i}s_{i}-{∥v_{i}^{*}\left(s_{i}\right)∥}^{2}\right)\right)-\sum_{i=1}^{n}\left(1-2\xi_{i}\right)\left(s_{i}^{T}Q_{i}s_{i}\right)\\ \; & -\sum_{i=1}^{n}\left(h_{i}\vartheta_{i}{\left(s_{j}\right)}^{2}-2\xi_{i}\sum_{j=1}^{m_{i}}∥v_{i}^{*}\left(s_{i}\right)∥b_{i,j}\vartheta_{i,j}\left(s_{j}\right)+\xi^{2}{∥v_{i}^{*}\left(s_{i}\right)∥}^{2}\right)+\varrho_{i}+\frac{1}{4}\sum_{i=1}^{n}G_{i,m}^{2}\delta_{i,m}^{2}, \end{array}$$
by denoting $\Lambda =diag\left\{h_{1},h_{2},\ldots ,h_{n}\right\}$ and $Z=[\vartheta_{1}\left(s_{1}\right),\ldots ,\vartheta_{n}\left(s_{n}\right),\;\xi_{1}∥v_{1}^{*}\left(s_{1}\right)∥,\ldots ,$ $\xi_{n}∥v_{n}^{*}\left(s_{n}\right)∥]$. Let the condition (Equation 31) be satisfied, so we have
(43)$$\begin{array}{c} {\dot{L}}_{i,1}\left(s\right)\leq -\sum_{i=1}^{n}\left(1-2\xi_{i}\right)\left(s_{i}^{T}Q_{i}s_{i}\right)-Z^{T}XZ+\varrho_{i}+\frac{1}{4}\sum_{i=1}^{n}G_{i,m}^{2}\delta_{i,m}^{2}, \end{array}$$
with $X=\left[\begin{array}{cc} \Lambda & A^{T}\\ A & I_{n} \end{array}\right]$ and $A=\left[\begin{array}{ccc} b_{11} & \cdots & b_{1n}\\ \vdots & \ddots & \vdots \\ b_{n1} & \cdots & b_{nn} \end{array}\right]$.From the matrix *X* expression, positive definiteness is maintained by choosing a sufficiently large Λ. In other words, there is $h_{i}^{*}>0$, such that $h_{i}>h_{i}^{*}$, ensuring $Z^{T}XZ>0$. Thus, the inequality (Equation 43) is further deduced as:
(44)$$\begin{array}{c} {\dot{L}}_{i,1}\left(s\right)\leq -\sum_{i=1}^{n}\left(1-2\xi_{i}\right)\lambda_{min}\left(Q_{i}\right){∥s_{i}∥}^{2}+\varrho_{i}+\frac{1}{4}\sum_{i=1}^{n}G_{i,m}^{2}\delta_{i,m}^{2}. \end{array}$$The inequality (Equation 44) means that ${\dot{L}}_{i,1}\left(s\right)<0$ whenever $s_{i}\left(t\right)$ lies outside the following set $N_{s_{i}}$:
(45)$$N_{s_{i}}=\left\{s_{i}:∥s_{i}∥\leq \sqrt{\frac{\frac{1}{4}G_{i,M}^{2}\delta_{i,M}^{2}+\varrho_{i}}{\lambda_{min}\left(Q_{i}\right)\left(1-2\xi_{i}\right)}}\right\}.$$Based on Lyapunov’s extension theorem [47], it is shown that the optimal performance functions $V_{i}^{*}\left(s_{i}\right)$ guarantee that the interconnected nonlinear system (Equation 15) with asymmetric input constraints is UUB. Since the performance function (Equation 7) and (Equation 22) yield the same results, it can be shown that the optimal performance function $J_{i}^{*}\left(x_{i}\right)$ guarantees that the interconnected nonlinear safety-critical system (1) with security constraints and asymmetric input constraints is UUB. □

## 4. Critic Network for Approximation

The critic neural network is introduced in this section, with the aim of approximating the optimal performance function. Then, the evaluation network of the auxiliary subsystem (Equation 21) is used to construct the estimated optimal control strategy. According to [48], $V_{i}^{*}\left(s_{i}\right)$ is expressed as: (46)$$V_{i}^{*}\left(s_{i}\right)=W_{c_{i}}^{T}\sigma_{c_{i}}\left(s_{i}\right)+\varepsilon_{c_{i}}\left(s_{i}\right),$$
where $\sigma_{c_{i}}\left(s_{i}\right)=\left[\sigma_{c_{i},1}\left(s_{i}\right),\sigma_{c_{i},2}\left(s_{i}\right),\ldots ,\sigma_{c_{i},N_{i}}\left(s_{i}\right)\right]\in R^{N_{i}}$ denotes the activation function, $W_{c_{i}}\in R^{N_{i}}$ denotes the ideal weight vector, $N_{i}$ denotes the number of neurons, and $\varepsilon_{c_{i}}\left(s_{i}\right)\in R^{N_{i}}$ is the reconstruction error of NN. The vector activation function $\sigma_{c_{i},p}\left(s_{i}\right)$ is denoted as a continuously differentiable function, where $p=1,2,\ldots ,N_{i}$. For $s_{i}\neq 0$, ${\left\{\sigma_{c_{i},p}\left(s_{i}\right)\right\}}_{p=1}^{N_{i}}$ is linearly independent. Then, the derivative of $V_{i}^{*}\left(s_{i}\right)$ can be expressed as: (47)$$\nabla V_{i}^{*}\left(s_{i}\right)=\nabla \sigma_{c_{i}}^{T}\left(s_{i}\right)W_{c_{i}}+\nabla \varepsilon_{c_{i}}\left(s_{i}\right),$$
where $\nabla \sigma_{c_{i}}\left(s_{i}\right)=\frac{\partial \sigma_{c_{i}}\left(s_{i}\right)}{\partial s_{i}}$ and $\nabla \varepsilon_{c_{i}}\left(s_{i}\right)=\frac{\partial \varepsilon_{c_{i}}\left(s_{i}\right)}{\partial s_{i}}$.

From Equations (Equation 27), (Equation 28) and (Equation 47), the optimal safety control policy $u_{i}^{*}\left(s_{i}\right)$ and the auxiliary control strategy $v_{i}^{*}\left(s_{i}\right)$ are rephrased as: (48)$$u_{i}^{*}\left(s_{i}\right)=-\lambda_{i}\tanh\left(\frac{1}{2\lambda_{i}}G_{i}^{T}\left(s_{i}\right)\nabla \sigma_{c_{i}}^{T}\left(s_{i}\right)W_{c_{i}}\right)+c_{d_{i}}+\varepsilon_{u_{i}}\left(s_{i}\right),$$
(49)$$v_{i}^{*}\left(s_{i}\right)=-\frac{1}{2\xi_{i}}{℘}_{i}^{T}\left(s_{i}\right)\nabla \sigma_{c_{i}}^{T}\left(s_{i}\right)W_{c_{i}}+\varepsilon_{v_{i}}\left(s_{i}\right),$$
where
$$\begin{array}{cc} \varepsilon_{u_{i}}\left(s_{i}\right) & =-\frac{1}{2}\left(I_{m_{i}}-\tanh^{2}\left(\zeta \right)\right)G_{i}^{T}\left(s_{i}\right)\nabla \varepsilon_{c_{i}}\left(s_{i}\right),\\ \varepsilon_{v_{i}}\left(s_{i}\right) & =-\frac{1}{2\xi_{i}}{℘}_{i}^{T}\left(s_{i}\right)\nabla \varepsilon_{c_{i}}\left(s_{i}\right), \end{array}$$
with $I_{m_{i}}={\left[1,1,\ldots ,1\right]}^{T}\in R^{m_{i}}$. The seclected value of ζ is between $\frac{1}{2\lambda_{i}}G_{i}^{T}\left(s_{i}\right)\nabla \sigma_{c_{i}}^{T}\left(s_{i}\right)W_{c_{i}}$ and $\frac{1}{2\lambda_{i}}G_{i}^{T}\left(s_{i}\right)\left(\nabla \sigma_{c_{i}}^{T}\left(s_{i}\right)W_{c_{i}}+\nabla \varepsilon_{c_{i}}\left(s_{i}\right)\right)$.

The ideal weight vector $W_{c_{i}}$ is not available and the optimal control strategy $u_{i}^{*}\left(s_{i}\right)$ is not directly applicable. Therefore, the estimated weight vector ${\hat{W}}_{c_{i}}$ is constructed to replace $W_{c_{i}}$ as: (50)$${\hat{V}}_{i}^{*}\left(s_{i}\right)={\hat{W}}_{c_{i}}^{T}\sigma_{c_{i}}\left(s_{i}\right).$$

The estimation error ${\tilde{W}}_{c_{i}}=W_{c_{i}}-{\hat{W}}_{c_{i}}$ is defined. Similarly, according to (Equation 50), the (Equation 49) and (Equation 48) are further developed as: (51)$${\hat{u}}_{i}\left(s_{i}\right)=-\lambda_{i}\tanh\left(\frac{1}{2\lambda_{i}}G_{i}^{T}\left(s_{i}\right)\nabla \sigma_{c_{i}}^{T}\left(s_{i}\right){\hat{W}}_{c_{i}}\right)+c_{d_{i}},$$
(52)$${\hat{v}}_{i}\left(s_{i}\right)=-\frac{1}{2\xi_{i}}{℘}_{i}^{T}\left(s_{i}\right)\nabla \sigma_{c_{i}}^{T}\left(s_{i}\right){\hat{W}}_{c_{i}}.$$

Combining (Equation 50), (Equation 51) and (Equation 52), the Hamiltonian is re-expressed as: (53)$$\begin{array}{cc} H(s_{i},{\hat{u}}_{i},{\hat{v}}_{i},\nabla {\hat{V}}_{i}\left(s_{i}\right))= & {\left(\nabla {\hat{V}}_{i}\left(s_{i}\right)\right)}^{T}\left(G_{i}^{T}\left(s_{i}\right){\hat{u}}_{i}+F_{i}\left(s_{i}\right)+{℘}_{i}\left(s_{i}\right){\hat{v}}_{i}\right)\\ \; & +\pi_{i}\left(s_{i}\right)+\gamma_{i}\left(s_{i},{\hat{u}}_{i},{\hat{v}}_{i}\right)-\alpha_{i}{\hat{V}}_{i}. \end{array}$$

According to (Equation 53), the error of the Hamiltonian is given by: (54)$$\begin{array}{cc} e_{i} & =H\left(s_{i},{\hat{u}}_{i},{\hat{v}}_{i},\nabla {\hat{V}}_{i}\left(s_{i}\right)\right)-H\left(s_{i},u_{i}^{*},v_{i}^{*},\nabla V_{i}^{*}\left(s_{i}\right)\right)\\ \; & =\pi_{i}\left(s_{i}\right)+s_{i}^{T}Q_{i}s_{i}+W_{i}\left({\hat{u}}_{i}\right)+\xi_{i}{\hat{v}}_{i}^{T}{\hat{v}}_{i}+{\hat{W}}_{c_{i}}^{T}\varrho_{i}, \end{array}$$
with $\varrho_{i}=\nabla \sigma_{c_{i}}\left(x_{i}\right)\left(G_{i}^{T}\left(s_{i}\right){\hat{u}}_{i}+F_{i}\left(s_{i}\right)+{℘}_{i}\left(s_{i}\right){\hat{v}}_{i}\right)-\alpha_{i}\sigma_{c_{i}}\left(s_{i}\right)$. In order to make $u_{i}\left(s_{i}\right)\to u_{i}^{*}\left(s_{i}\right)$, the error $e_{i}$ should be guaranteed to be sufficiently small. To solve this issue, a critic weight adjustment law ${\hat{W}}_{c_{i}}$ is proposed to minimize the objective function $\phi_{i}=\frac{1}{2}e_{i}^{T}e_{i}$. Next, the critic updating law is developed as: (55)$${\hat{W}}_{c_{i}}=-\frac{\alpha_{c_{i}}\varrho_{i}e_{i}}{{\left(1+\varrho_{i}^{T}\varrho_{i}\right)}^{2}},$$
where the constant $\alpha_{c_{i}}$ is the positive learning rate.

**Remark** **5.**
*To minimize the Hamiltonian error $e_{i}$, it is necessary to maintain the derivative of $\phi_{i}$ as ${\dot{\phi }}_{i}<0$. Therefore, the critic weight adjustment law is derived by employing the normalization term ${\left(1+\varrho_{i}^{T}\varrho_{i}\right)}^{-2}$ and applying the gradient descent method with respect to ${\hat{W}}_{c_{i}}$ [49].*


By considering the definition of ${\tilde{W}}_{c_{i}}$, we obtain
(56)$${\dot{\tilde{W}}}_{c_{i}}=-\alpha_{c_{i}}\ell_{i}\ell_{i}^{T}{\tilde{W}}_{c_{i}}+\frac{\alpha_{c_{i}}\ell_{i}e_{H_{i}}}{{𝚤}_{i}},$$
where $\ell_{i}=\frac{\varrho_{i}}{1+\varrho_{i}^{T}\varrho_{i}}$ and ${𝚤}_{i}=1+\varrho_{i}^{T}\varrho_{i}$. $e_{H_{i}}$ denotes the residual error, defined as $e_{H_{i}}=\nabla \sigma_{c_{i}}\left(x_{i}\right)\left(G_{i}^{T}\left(s_{i}\right){\hat{u}}_{i}+F_{i}\left(s_{i}\right)+{℘}_{i}\left(s_{i}\right){\hat{v}}_{i}\right)$.

The proposed decentralized DSC strategy for the ith subsystem with a single critic-NN is illustrated in Figure 1.

## 5. Stability Analysis

This section focuses on the stability of the *n*-auxiliary subsystem for the given control scheme. We need to make some Assumptions to satisfy the theorem.

**Assumption** **4.**
*For $s_{i}\in \Omega_{i},\;i=1,\ldots ,n$, there exist some positive constants $D_{\varepsilon_{u_{i}}},\eta_{i,M},\;D_{\sigma_{c_{i}}},\;D_{\varepsilon_{v_{i}}}$ and $D_{e_{H_{i}}}$ satisfying $∥\varepsilon_{u_{i}}\left(s_{i}\right)∥\leq D_{\varepsilon_{u_{i}}}$, $∥{℘}_{i}\left(s_{i}\right)∥\leq {℘}_{i,M}$, $∥\nabla \sigma_{c_{i}}\left(s_{i}\right)∥\leq D_{\sigma_{c_{i}}},∥\varepsilon_{v_{i}}\left(s_{i}\right)∥\leq D_{\varepsilon_{v_{i}}}$ and $∥e_{H_{i}}∥\leq D_{e_{H_{i}}}$.*


**Assumption** **5.**
*Consider the time period $\left[t,t+t_{k}\right]$ and $t_{k}>0$. Then, the term $\ell_{i}\ell_{i}^{T}$ fulfills the following condition:*

(57)
$$\epsilon_{i}I_{N_{i}}\leq \ell_{i}\ell_{i}^{T}\leq s_{i}I_{N_{i}},$$

*where $\epsilon_{i}$ and $s_{i}$ are positive constants.*


**Theorem** **2.**
*For the nonlinear interconnected safety-critical system (Equation 15), we design the estimated optimal safety policies and auxiliary control strategies as (Equation 51) and (Equation 52), respectively. Assume that Assumptions 1–5 hold. If ${\hat{W}}_{c_{i}}$ is updated by (Equation 55), then $s_{i}$ and ${\hat{W}}_{c_{i}}$ are UUB if $\alpha_{c_{i}}$ in (Equation 55) satisfies*

(58)
$$\alpha_{c_{i}}>\frac{{℘}_{i,M}^{2}D_{\sigma_{c_{i}}}^{2}}{\xi_{i}\lambda_{min}\left(\ell_{i}\ell_{i}^{T}\right)}.$$



**Proof.** The candidate Lyapunov function is considered to be:
(59)$$L_{i}\left(t\right)=\sum_{i=1}^{n}\left(V_{i}^{*}\left(s_{i}\right)+\frac{1}{2}{\tilde{W}}_{c_{i}}^{T}{\tilde{W}}_{c_{i}}\right).$$Then, defining $L_{i,1}\left(t\right)=V_{i}^{*}\left(s_{i}\right)$ and $L_{i,2}\left(t\right)=\frac{1}{2}{\tilde{W}}_{c_{i}}^{T}{\tilde{W}}_{c_{i}}$, the time derivative by $L_{i,1}\left(t\right)$ is
(60)$$\begin{array}{cc} {\dot{L}}_{i,1}\left(t\right)= & {\left(\nabla V_{i}^{*}\left(s_{i}\right)\right)}^{T}\left(G_{i}^{T}\left(s_{i}\right){\hat{u}}_{i}+F_{i}\left(s_{i}\right)+{℘}_{i}\left(s_{i}\right){\hat{v}}_{i}\right)\\ = & {\left(\nabla V_{i}^{*}\left(s_{i}\right)\right)}^{T}\left(G_{i}^{T}\left(s_{i}\right)u_{i}^{*}+F_{i}\left(s_{i}\right)+{℘}_{i}\left(s_{i}\right)v_{i}^{*}\right)\\ \; & +\underset{\beta_{3}}{\underbrace{{\left(\nabla V_{i}^{*}\left(s_{i}\right)\right)}^{T}G_{i}^{T}\left(s_{i}\right)\left({\hat{u}}_{i}-u_{i}^{*}\right)}}+\underset{\beta_{4}}{\underbrace{{\left(\nabla V_{i}^{*}\left(s_{i}\right)\right)}^{T}{℘}_{i}\left(s_{i}\right)\left({\hat{v}}_{i}-v_{i}^{*}\right)}}. \end{array}$$Combining (Equation 29), (Equation 34) and (Equation 35). The (Equation 60) is further deduced as:
(61)$$\begin{array}{cc} {\dot{L}}_{i,1}\left(t\right)= & \alpha_{i}V_{i}^{*}-\pi_{i}\left(s_{i}\right)-s_{i}^{T}Q_{i}s_{i}-W_{i}\left(u_{i}^{*}\right)+\xi_{i}{∥v_{i}^{*}\left(s_{i}\right)∥}^{2}+\beta_{3}+\beta_{4}. \end{array}$$According to Lemma 1, and taking into account (Equation 40), (Equation 48), (Equation 51), we observe that the $\beta_{3}$ term in (Equation 61) is satisfied by
(62)$$\begin{array}{cc} \beta_{3} & \leq \lambda_{i}^{2}∥\tanh^{-1}\left(\frac{u_{i}^{*}\left(s_{i}\right)-c_{d_{i}}}{\lambda_{i}}\right)∥+{∥{\hat{u}}_{i}-u_{i}^{*}∥}^{2}\\ \; & \leq \beta_{1}+\underset{\beta_{5}}{\underbrace{{∥\lambda_{i}\left(\tanh\left(Y_{i,1}\left(s_{i}\right)\right)-\tanh\left(Y_{i,2}\left(s_{i}\right)\right)\right)-\varepsilon_{u_{i}}\left(s_{i}\right)∥}^{2}}}\\ \; & \leq \frac{1}{4}G_{i,M}^{2}\delta_{i,M}^{2}+\beta_{5}, \end{array}$$
where $Y_{i}\left(s_{i}\right)=\frac{1}{2\lambda_{i}}G_{i}^{T}\left(s_{i}\right)\nabla V_{i}^{*}\left(s_{i}\right)$. Then, based on the fact $∥\tanh(Y_{i,k}\left(s_{i}\right))∥\leq \sqrt{m_{i}},$ k=1,2 in [44], according to Assumption 5, $\beta_{5}$ is derived as:
(63)$$\begin{array}{cc} \beta_{5} & \leq 2\lambda_{i}^{2}{∥\tanh\left(Y_{i,1}\left(s_{i}\right)\right)-\tanh\left(Y_{i,2}\left(s_{i}\right)\right)∥}^{2}+2{∥\varepsilon_{u_{i}}\left(s_{i}\right)∥}^{2}\\ \; & \leq 4\lambda_{i}^{2}\left({∥\tanh(Y_{i,1}\left(s_{i}\right))∥}^{2}+{∥\tanh(Y_{i,2}\left(s_{i}\right))∥}^{2}\right)\;+2{∥\varepsilon_{u_{i}}\left(s_{i}\right)∥}^{2}\\ \; & \leq 8\lambda_{i}^{2}m_{i}+2D_{\varepsilon_{u_{i}}}^{2}. \end{array}$$Similarly, the last term of (Equation 61) is deduced from (Equation 35), (Equation 49) and (Equation 52) as:
(64)$$\begin{array}{cc} \beta_{4} & \leq -\xi_{i}{∥v_{i}^{*}∥}^{2}-\xi_{i}{∥{\hat{v}}_{i}-v_{i}^{*}∥}^{2}\\ \; & \leq -\xi_{i}{∥v_{i}^{*}∥}^{2}+2\xi_{i}\left({∥{\hat{v}}_{i}∥}^{2}-{∥v_{i}^{*}∥}^{2}\right)+2\xi_{i}{∥\varepsilon_{v_{i}}∥}^{2}\\ \; & \leq -\xi_{i}{∥v_{i}^{*}∥}^{2}+\frac{1}{2\xi_{i}}{℘}_{i,M}^{2}D_{\sigma_{c_{i}}}^{2}{∥{\tilde{W}}_{c_{i}}∥}^{2}+2\xi_{i}D_{\varepsilon_{v_{i}}}^{2}. \end{array}$$By using (Equation 38), (Equation 62)–(Equation 64) and the fact that $∥\alpha_{i}V_{i}^{*}\left(s_{i}\right)∥\leq \varrho_{i,M}$ the following is derived:
(65)$$\begin{array}{c} {\dot{L}}_{i,1}\left(t\right)\leq -\lambda_{min}\left(Q_{i}\right){∥s_{{}_{i}}∥}^{2}+\frac{1}{2\xi_{i}}{℘}_{i,M}^{2}D_{\sigma_{c_{i}}}^{2}{∥{\tilde{W}}_{c_{i}}∥}^{2}+\Theta_{i}, \end{array}$$
with $\Theta_{i}=\varrho_{i,M}+\frac{1}{2}G_{i,M}^{2}\delta_{i,M}^{2}+8\lambda_{i}^{2}m_{i}+2D_{\varepsilon_{u_{i}}}^{2}+2\xi_{i}D_{\varepsilon_{v_{i}}}^{2}$.The error weight update law ${\tilde{W}}_{c_{i}}$. $L_{i,1}\left(t\right)$ is considered with the time derivative
(66)$${\dot{L}}_{i,2}\left(t\right)=-\alpha_{c_{i}}{\tilde{W}}_{c_{i}}^{T}\ell_{i}\ell_{i}^{T}{\tilde{W}}_{c_{i}}+\alpha_{c_{i}}\frac{{\tilde{W}}_{c_{i}}^{T}\ell_{i}}{{𝚤}_{i}}e_{H_{i}}.$$Combining Lemma 1 and Assumption 4, the following conclusion is drawn:
(67)$$\alpha_{c_{i}}\frac{{\tilde{W}}_{c_{i}}^{T}\ell_{i}}{{𝚤}_{i}}e_{H_{i}}\leq \frac{\alpha_{c_{i}}}{2}{\tilde{W}}_{c_{i}}^{T}\ell_{i}\ell_{i}^{T}{\tilde{W}}_{c_{i}}+\frac{\alpha_{c_{i}}}{2}D_{e_{H_{i}}}^{2}.$$Combining inequalities (Equation 66) and (Equation 67), we derive the following inequalities:
(68)$${\dot{L}}_{i,2}\left(t\right)\leq -\frac{\alpha_{c_{i}}}{2}\lambda_{min}\left(\ell_{i}\ell_{i}^{T}\right){∥{\tilde{W}}_{c_{i}}∥}^{2}+\frac{\alpha_{c_{i}}}{2}D_{e_{H_{i}}}^{2}.$$Substituting (Equation 65) and (Equation 68) into (Equation 59), the following inequality is obtained:
(69)$${\dot{L}}_{i}\left(t\right)\leq \sum_{i=1}^{n}\left(-\lambda_{min}\left(Q_{i}\right){∥s_{i}∥}^{2}-x_{i}{∥{\tilde{W}}_{c_{i}}∥}^{2}+\Theta_{i}+\frac{\alpha_{c_{i}}}{2}D_{e_{H_{i}}}^{2}\right),$$
where $x_{i}=\frac{\alpha_{c_{i}}}{2}\lambda_{min}\left(\ell_{i}\ell_{i}^{T}\right)-\frac{1}{2\xi_{i}}{℘}_{i,M}^{2}D_{\sigma_{c_{i}}}^{2}$, $\lambda_{min}\left(\ell_{i}\ell_{i}^{T}\right)$ means the minimum eigenvalue of $\ell_{i}\ell_{i}^{T}$.Therefore, Equations (Equation 58) and (Equation 69) mean ${\dot{L}}_{i}\left(t\right)<0$, provided that the parameters $s_{i}$ and ${\tilde{W}}_{c_{i}}$ are not in the set of
(70)$$N_{i}\left\{s_{i}:∥s_{i}∥\leq \sqrt{\frac{2\Theta_{i}+D_{e_{H_{i}}}^{2}}{2\lambda_{min}\left(Q_{i}\right)}}\right\},$$
(71)$$N_{{\tilde{W}}_{c_{i}}}\left\{{\tilde{W}}_{c_{i}}:∥{\tilde{W}}_{c_{i}}∥\leq \sqrt{\frac{2\Theta_{i}+D_{e_{H_{i}}}^{2}}{x_{i}}}\right\}.$$Introducing Lyapunov’s extension theorem, ref. [47], ensures the stability of the closed-loop system. This proof ensures that the weight estimation error ${\tilde{W}}_{c_{i}}$ is UUB. At this point, this completes the proof process. □

**Remark** **6.**
*In contrast to techniques that aim to achieve input saturation [10,13], this article proposes an RL technique to solve the optimal DSC problem with safety constraints and asymmetric input constraints. This approach ensures not only the safety of the system but also minimizes the input constraints. Therefore, the developed reinforcement learning technique, based on security constraints and asymmetric input constraints, is better suited for some project applications, particularly for systems where the system state must be globally within the security settings.*


## 6. Simulation Example

In this section, we provide a simulation example to verify the effectiveness of the proposed approach. The simulation involved a dual-linked robotic arm system [42]. The state space model of the system is defined by
(72)$$\begin{array}{cc} {\dot{x}}_{1,1} & =x_{1,2},\\ {\dot{x}}_{1,2} & =-\frac{M_{1}}{{\tilde{G}}_{1}}x_{1,2}-\frac{m_{1}\tilde{g}{\tilde{l}}_{1}}{{\tilde{G}}_{1}}\sin\left(x_{1,1}\right)+\frac{1}{{\tilde{G}}_{1}}u_{1}+△h_{1},\\ {\dot{x}}_{2,1} & =x_{2,2},\\ {\dot{x}}_{2,2} & =-\frac{M_{2}}{{\tilde{G}}_{2}}x_{2,2}-\frac{m_{2}\tilde{g}{\tilde{l}}_{2}}{{\tilde{G}}_{2}}\sin\left(x_{2,1}\right)+\frac{1}{{\tilde{G}}_{2}}u_{2}+△h_{2}, \end{array}$$
where $x_{i,1}$ and $x_{i,2}$(i=1,2) indicate the angular location of the robot arm, $u_{i}$ stands for control input, and the $△h_{i}=\eta_{i}P_{i}$ represents the interconnection terms. The other parameters of the robotic arm system (Equation 72) are depicted in Table 1. The initial system state was selected as $x_{0}={\left[2,2,2,2\right]}^{T}$. We first defined the state variable $x_{i}={\left[x_{i,1},x_{i,2}\right]}^{T}$ and constructed the internal dynamics and input gain matrix as follows: $$f_{i}\left(x_{i}\right)=\left[\begin{array}{c} x_{i,2}\\ -\frac{M_{i}}{{\tilde{G}}_{i}}x_{i,2}-\frac{m_{i}\tilde{g}{\tilde{l}}_{i}}{{\tilde{G}}_{i}}\sin\left(x_{i,1}\right) \end{array}\right]+\left[\begin{array}{c} 0\\ \frac{1}{{\tilde{G}}_{i}} \end{array}\right]u_{i}+\left[\begin{array}{c} 1\\ 0 \end{array}\right]P_{i}\left(x_{i}\right),$$
where $P_{1}\left(x_{1}\right)$, $P_{2}\left(x_{2}\right)$ denote the uncertain interconnection terms of subsystems 1 and 2, i.e.,
$$\begin{array}{cc} P_{1}\left(x_{1}\right) & =0.1x_{1,1}\sin\left(x_{2,2}\right),\\ P_{2}\left(x_{2}\right) & =(x_{1,2}-3\sin\left(0.1x_{2,1}\right)). \end{array}$$

Furthermore, the two robotic arm subsystems were in a state that satisfied the below security constraints: (73)$$\begin{array}{cc} \; & x_{1,1}\in \left(-0.5,2.9\right),\;x_{1,2}\in \left(-1.5,2.5\right),\\ \; & x_{2,1}\in \left(-1,2.5\right),\;x_{2,2}\in \left(-3.5,3\right). \end{array}$$

Therefore, to deal with the security constraint, the following system of transformations without security constraint was obtained, using the BF-based system transformation (Equation 13): (74)$$s_{i}=F_{i}\left(s_{i}\right)+G_{i}\left(s_{i}\right)u_{i}+{℘}_{i}\left(s_{i}\right)U_{i},$$
where
(75)$$\begin{array}{cc} F_{i}\left(s_{i}\right) & =\left[\begin{array}{c} \frac{a_{i,2}A_{i,2}\left(e^{\frac{s_{i,2}}{2}}-e^{-\frac{s_{i,2}}{2}}\right)}{a_{i,2}e^{\frac{s_{i,2}}{2}}-A_{i,2}e^{-\frac{s_{i,2}}{2}}}\frac{a_{i,1}^{2}e^{s_{i,1}}-2a_{i,1}A_{i,1}+A_{i,1}e^{-s_{i,1}}}{A_{i,1}a_{i,1}^{2}-a_{1}A_{i,1}^{2}}\\ f_{i}\left(B^{-1}\left(s_{i}\right)\right)\frac{a_{i,2}^{2}e^{s_{i,2}}-2a_{i,2}A_{i,2}+A_{i,2}e^{-s_{i,2}}}{A_{i,2}a_{i,2}^{2}-a_{i,2}A_{i,2}^{2}} \end{array}\right],\\ G_{i}\left(s_{i}\right) & =\left[\begin{array}{c} 0\\ \frac{1}{\tilde{G}}\frac{a_{i,2}^{2}e^{s_{i,2}}-2a_{i,2}A_{i,2}+A_{i,2}e^{-s_{i,2}}}{A_{i,2}a_{i,2}^{2}-a_{2}A_{i,2}^{2}} \end{array}\right],\\ {℘}_{i}\left(s_{i}\right) & =\left[\begin{array}{c} \frac{a_{i,2}^{2}e^{s_{i,2}}-2a_{i,2}A_{i,2}+A_{i,2}e^{-s_{i,2}}}{A_{i,2}a_{i,2}^{2}-a_{2}A_{i,2}^{2}}\\ 0 \end{array}\right]. \end{array}$$

For the transformed dual-linked robotic arm system (Equation 74), the initial state was chosen by $s_{i,0}={\left[s_{i,0}\left(1\right),s_{i,0}\left(2\right)\right]}^{T}={\left[B\left(x_{i,0}\left(1\right);a_{i,1},A_{i,1}\right),B\left(x_{i,0}\left(2\right);a_{i,2},A_{i,2}\right)\right]}^{T}$. The discount factors were chosen as $\alpha_{1}=1$ and $\alpha_{2}=0.1$. The matrices were designed as $Q_{1}=0.5I^{2}$ and $Q_{2}=I^{2}$, $R_{1}=1$ and $R_{2}=1$. The upper and lower limits were allocated as below: $h_{1\;max}=0.75$, $h_{1\;min}=-0.25$ and $h_{2\;max}=1.5$, $h_{2\;min}=-0.5$. Let $\vartheta_{1}=∥s_{1}∥$ and $\vartheta_{2}=∥s_{2}∥$. Additional design factors were setup as below: $\xi_{1}=8,\;\xi_{2}=4,\;a_{c_{1}}=2,\;a_{c_{2}}=2$. Choose the activation functions $\sigma_{c_{i}}\left(s_{i}\right)={\left[s_{1,1}^{2},s_{1,1}s_{1,2},s_{1,2}^{2}\right]}^{T}$ and $\sigma_{c_{i}}\left(s_{i}\right)={\left[s_{2,1}^{2},s_{2,1}s_{2,2},s_{2,2}^{2}\right]}^{T}$.

The simulation outcomes are presented in Figure 2, Figure 3, Figure 4, Figure 5, Figure 6, Figure 7, Figure 8, Figure 9, Figure 10, Figure 11, Figure 12 and Figure 13. The states of the system are depicted in Figure 2 and Figure 8, and it can be observed that the closed-loop system stabilized after 20 s and 35 s, respectively. However, the system failed to meet the specified security constraints. Figure 3 and Figure 9, shown in comparison with Figure 2 and Figure 8, not only assured that the system states converged to zero, but also satisfied the given safety constraints. The evolving states $s_{1}\left(t\right)$ and $s_{2}\left(t\right)$ are presented in Figure 4 and Figure 10, based on the safe control method with asymmetric input constraints. The optimal DSC policies are shown in Figure 5 and Figure 11. We found that the optimal DSC policies were restricted to the asymmetric set [−0.25,0.75] and [−0.5,1.5]. Figure 6 and Figure 12 represent the optimal auxiliary control strategies for subsystems 1 and 2, respectively. Figure 7 and Figure 13 show the critic updated laws. It can be observed that the weights converged after 15 s. According to Theorem 3, we concluded that the proposed optimal safety control policy and the auxiliary control policy could stabilize the closed-loop nonlinear system and satisfy the safety constraints on the system state. Moreover, the optimal control policy eventually converged to a predefined set of constraints. Finally, the results of the simulation showed that the presented optimal DSC solution for constrained interconnected nonlinear safety-critical systems, affected by system state constraints, is effective.

## 7. Conclusions

This article presents an RL-based DSC scheme for interconnected nonlinear safety-critical systems with security constraints and asymmetric input constraints. The proposed method transformed an interconnected nonlinear safety-critical system with security and asymmetric input constraints into an interconnected nonlinear safety-critical system with only asymmetric input constraints by using the barrier function. The non-quadratic utility function was added to the performance function to address the asymmetric input constraint. The critic network was also used to approach the optimal performance function and to establish the best security policy. Our control scheme stabilizes the closed-loop system and minimizes the improved performance function. In addition, the simulation results demonstrated the efficacy of the proposed distributed security solution. Future work will explore the optimal safety control of stochastic interconnected nonlinear systems with event triggering.

## Figures and Tables

**Figure 1 entropy-25-01158-f001:**
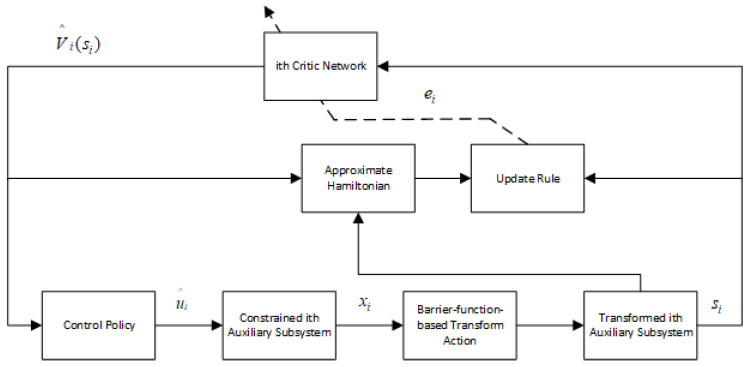
The block diagram of the developed optimal DSC scheme.

**Figure 2 entropy-25-01158-f002:**
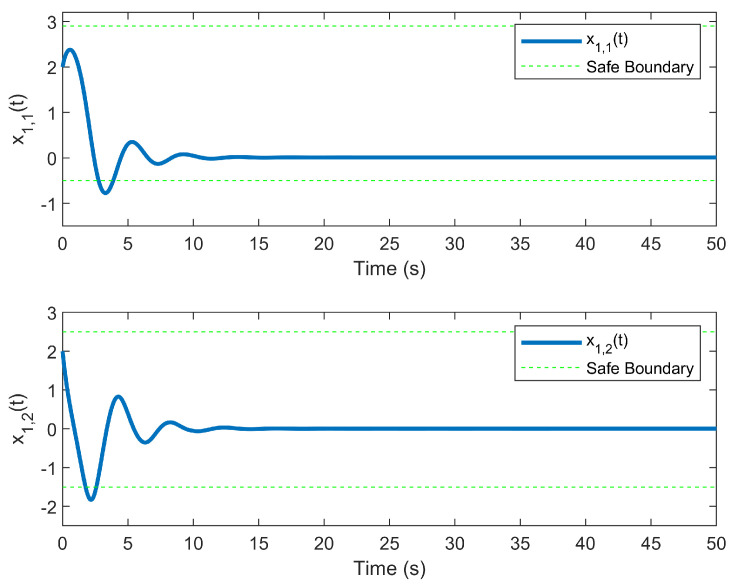
Evolution of state $x_{1}\left(t\right)$ without using the DSC method.

**Figure 3 entropy-25-01158-f003:**
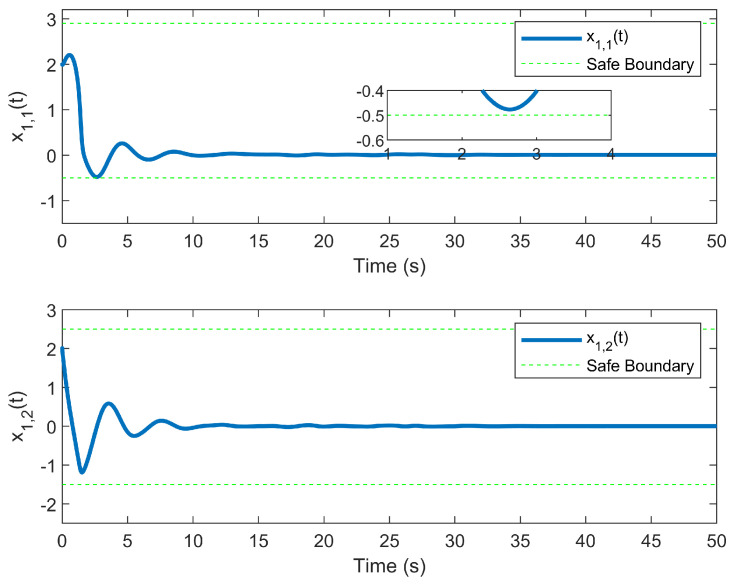
Evolution of state $x_{1}\left(t\right)$ using the DSC method.

**Figure 4 entropy-25-01158-f004:**
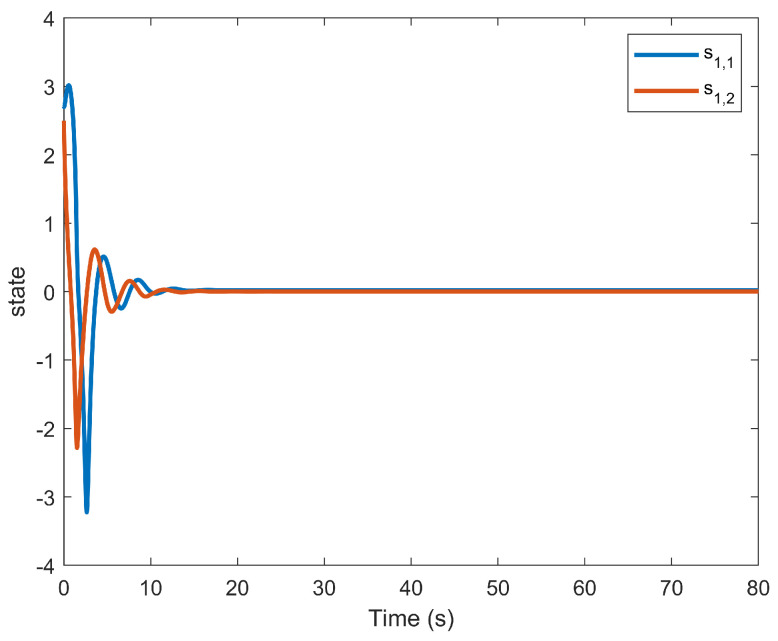
Evolution of state $s_{1}\left(t\right)$ using the DSC method.

**Figure 5 entropy-25-01158-f005:**
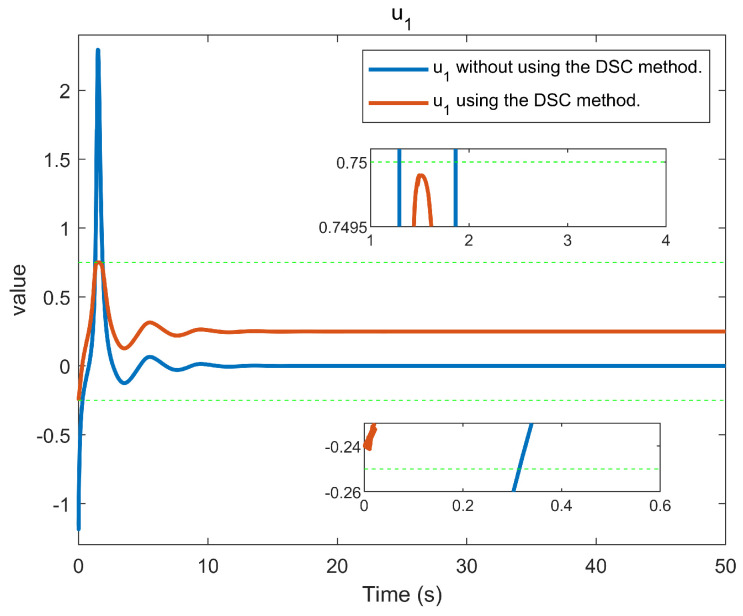
Control evolution of input $u_{1}$.

**Figure 6 entropy-25-01158-f006:**
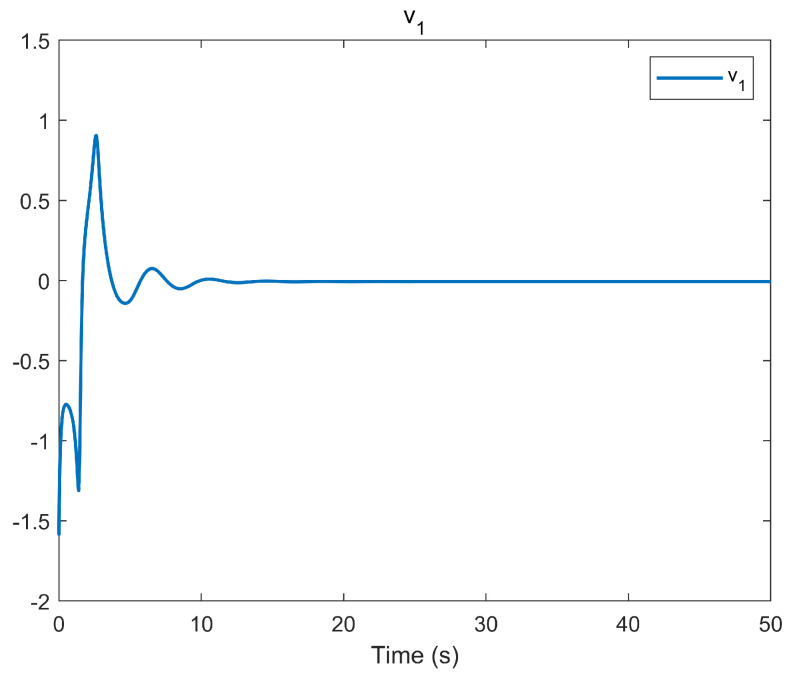
Evolution of the auxiliary control input $v_{1}$ using the DSC method.

**Figure 7 entropy-25-01158-f007:**
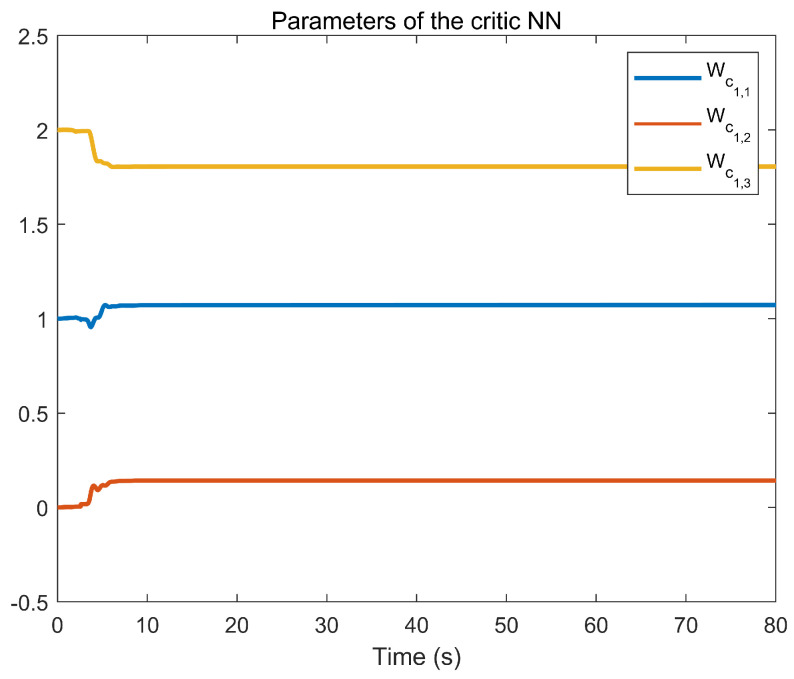
Evolution of the critic weight vector $W_{c_{1}}$ using the DSC method.

**Figure 8 entropy-25-01158-f008:**
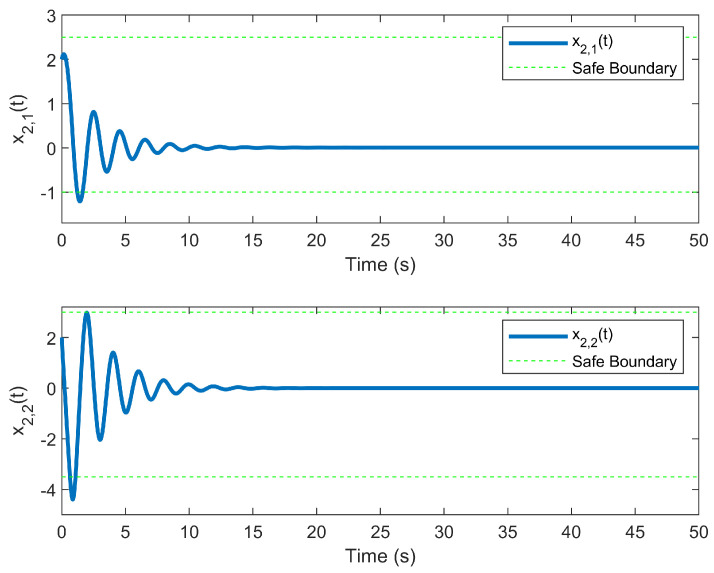
Evolution of state $x_{2}\left(t\right)$ without using the DSC method.

**Figure 9 entropy-25-01158-f009:**
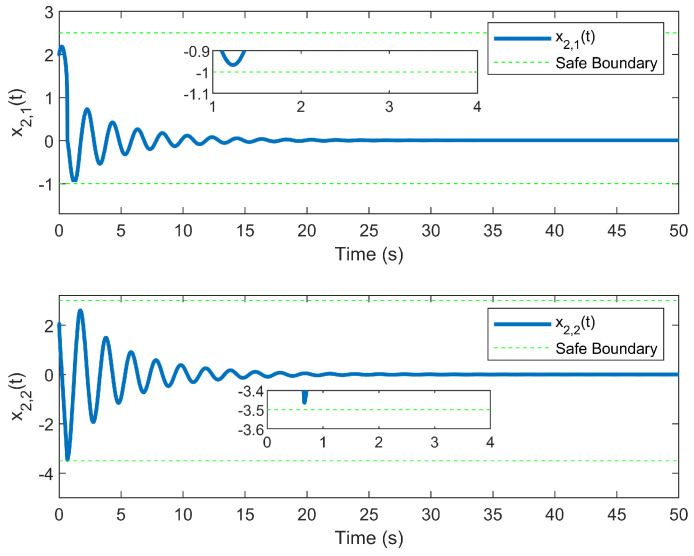
Evolution of state $x_{2}\left(t\right)$ using the DSC method.

**Figure 10 entropy-25-01158-f010:**
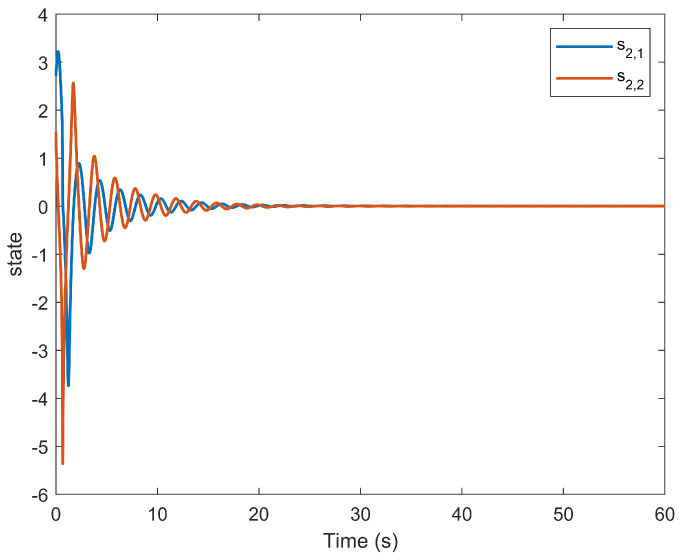
Evolution of state $s_{2}\left(t\right)$ using the DSC method.

**Figure 11 entropy-25-01158-f011:**
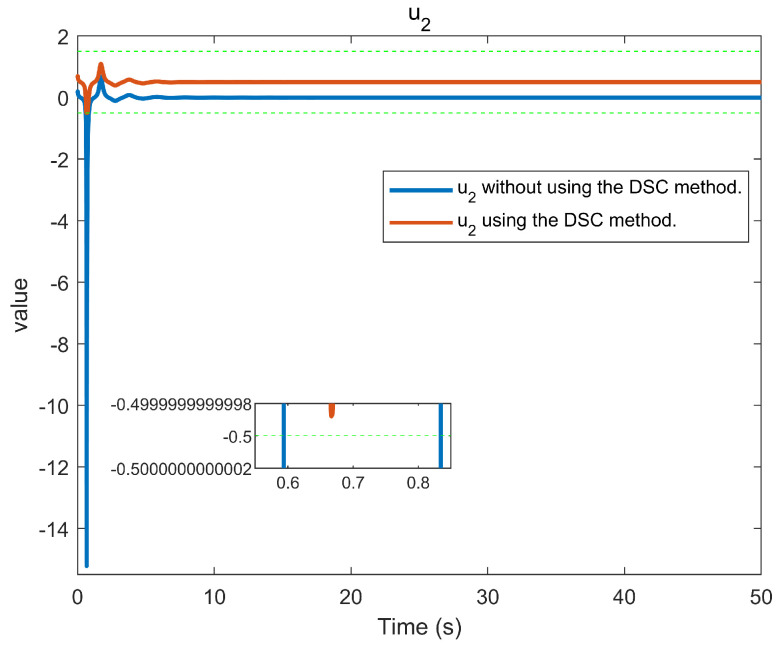
Control evolution of input $u_{2}$.

**Figure 12 entropy-25-01158-f012:**
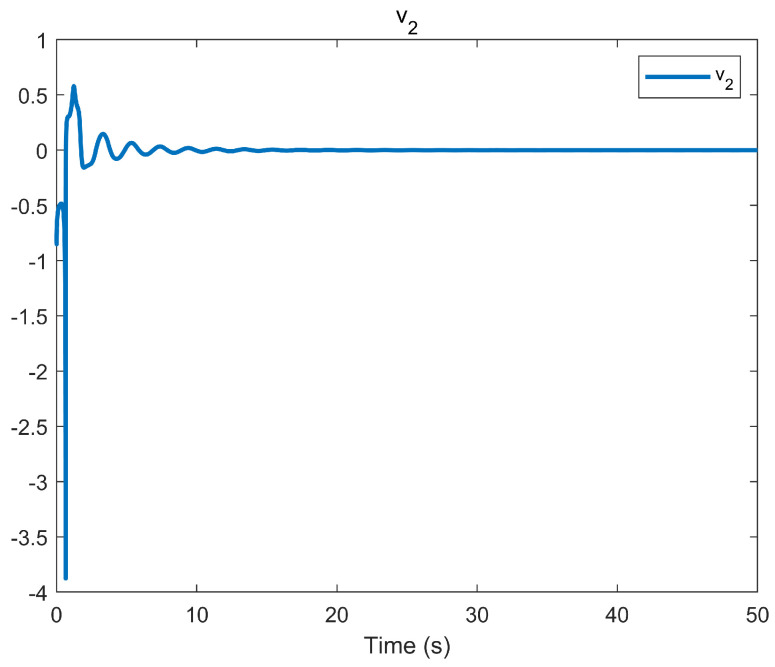
Evolution of the auxiliary control input $v_{2}$ using the DSC method.

**Figure 13 entropy-25-01158-f013:**
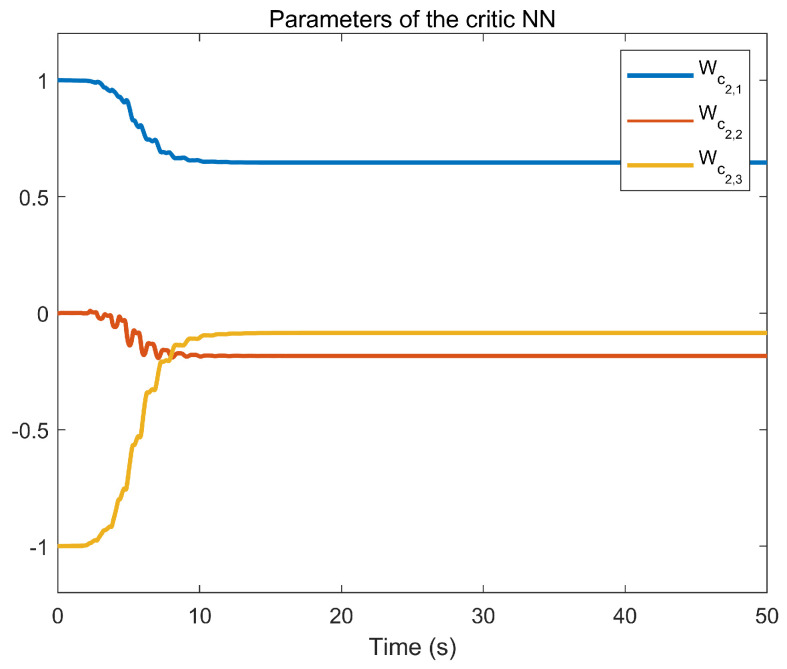
Evolution of the critic weight vector $W_{c_{2}}$ using the DSC method.

**Table 1 entropy-25-01158-t001:** Meanings and values of symbols used in robotic arm systems.

The *i*th Subsystem	Parameter	Meaning	Value
	$m_{1}$	Mass of payload	5 kg
	$M_{1}$	Viscous friction	2 N
The first subsystem	${\tilde{l}}_{1}$	Length of the arm	0.5 m
	${\tilde{G}}_{1}$	Moment of inertia	10 kg
	${\tilde{g}}_{1}$	Acceleration of gravity	9.81 m/s
	$m_{2}$	Mass of payload	10 kg
	$M_{2}$	Viscous friction	2 N
The second subsystem	${\tilde{l}}_{2}$	Length of the arm	1 m
	${\tilde{G}}_{2}$	Moment of inertia	10 kg
	${\tilde{g}}_{2}$	Acceleration of gravity	9.81 m/s

## Data Availability

The authors can confirm that all relevant data are included in the article.

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
