# Peer review of "Reinforcement Learning-Based Decentralized Safety Control for Constrained Interconnected Nonlinear Safety-Critical Systems"

_entropy, 2023, doi:10.3390/e25081158_

Round 1

Reviewer 1 Report

See the attachment.

None.

Reviewer 2 Report

1. According to Line 362, Remark 6, the authors proposed a technique based on RL to solve DSC. However, essential elements such as RL's markov decision property, bellman equation, Q-learning, and monte carlo are not presented. There isn't enough information on how it is RL-based one. Proposal for technology based on RL is not detailed, so more informatin is required.

2. In the introduction section, the authors proposed a RL based DSC to overcome the shortcomings in references [9]-[13]. The most important difference between the aforementioned papers and the author's paper is presented as "However, the above [9–13] mainly focused on a single continuous-time/ discrete-time nonlinear optimal security control. The security control of interconnected systems has not been fully resolved.". 

However, the method to overcome these disadvantages was not explained in detail in the contribution description (lines 84-97). A more specific explanation should be added to that section.

3. Comparative studies and discussions with other DSC methods may also be added.

4. The statement to support that there are numerous control constraints on the 65th line lacks reasonable evidence. It appears that there is a need for relevant references.

5. In equations 2 and 3, there is no explanation provided for the function P_i(x). Additional clarification is required regarding the nature of this function.

6. Line 135, it is necessary to specify the exact target or condition instead of using the expression "above condition."

The manuscript seems to be fine. But to improve the quality, it is recommended to take the grammar check-up and language polishing. 

Reviewer 3 Report

The authors propose a decentralized safety control of constrained interconnected nonlinear systems, using a strategy based on the reinforcement learning theory. The concept appears very interesting, but some doubts arise from the revision. The authors are invited to answer all the following comments.

1.  The authors should better explain the innovations of the proposed approach, since there are many works on Barrier Functions, constraints, etc.

2. The authors should improve the technical literature with some recent works (see for example, DOI: 10.1109/TAC.2022.3197562, https://doi.org/10.1146/annurev-control-042920-020211, DOI: 10.1109/LRA.2021.3070252, DOI: 10.1109/ACCESS.2022.3150926, https://doi.org/10.1016/j.epsr.2022.108609 and the references therein) .

3. The authors should also explain the term "decentralized" used for the control approach, and the meaning that they give to it.

4. The UUB stability could be worrying, especially in real applications. Is it possible to extend the results to asymptotic stability? (see for example the interesting results in DOI: 10.1109/TAC.2022.3197562). Otherwise, it could be interesting a sensitivity analysis on the magnitude of the residual set of convergence, in connection with the control tasks.

5. Is it possible to reduce the settling time in the simulation scenario?

6. At line 158, page 5, there is a wrong formatting for the couple (a,A). A careful rereading is recommended.

7.  It is mandatory a simulative comparison analysis with some decentralized approaches proposed in the technical literature, in order to better prove the advantages of your proposal

Minor editing are required for the English language.

Round 2

Reviewer 2 Report

All of my comments have been well addressed, and the paper can be accepted.

The manuscript seems to be fine. However, some sentences and expressions still need to be checked again.

Reviewer 3 Report

The authors replied to all my questions. Accordingly, the paper can be accepted now.